

# Introduction to Special Issue - In-depth study of air pollution sources and processes within Beijing and its surrounding region (APHH-Beijing)

Zongbo Shi[1,2*], Tuan Vu[1], Simone Kotthaus[3,4], Sue Grimmond[3], Roy M. Harrison[1†], Siyao Yue[5], Tong Zhu[6], James Lee[7,8], Yiqun Han[6,9], Matthias Demuzere[10], Rachel E Dunmore[7], Lujie Ren[2,5], Di Liu[1], Yuanlin Wang[5,11], Oliver Wild[11], James Allan[12,13], Janet Barlow[3], David Beddows[1], William J, Bloss[1], David Carruthers[14], David C Carslaw[7,15], Lia Chatzidiakou[16], Leigh Crilley[1], Hugh Coe[12], Tie Dai[5], Ruth Doherty[17], Fengkui Duan[18], Pingqing Fu[2,5], Baozhu Ge[5], Maofa Ge[19], Daobo Guan[20], Jacqueline F. Hamilton[7], Kebin He[18], Matthew Heal[17], Dwayne Heard[21], C Nicholas Hewitt[11], Min Hu[6], Dongsheng Ji[5], Xujiang Jiang[18], Rod Jones[16], Markus Kalberer[16,a], Frank J Kelly[9], Louisa Kramer[1], Ben Langford[22], Chun Lin[17], Alastair C Lewis[7], Jie Li[5], Weijun Li[23], Huan Liu[18], Miranda Loh[24], Keding Lu[6], Graham Mann[25], Gordon McFiggans[12], Mark Miller[26], Graham Mills[27], Paul Monk[28], Eiko Nemitz[22], Fionna O'Connor[29], Bin Ouyang[11,16], Paul I. Palmer[17], Carl Percival[12,b], Olalekan Popoola[16], Claire Reeves[27], Andrew R Rickard[7,8], Longyi Shao[30], Guangyu Shi[5], Dominick Spracklen[25], David Stevenson[17], Yele Sun[5], Zhiwei Sun[31], Shu Tao[32], Shengrui Tong[19], Qingqing Wang[5], Wenhua Wang[30], Xinming Wang[33], Zifang Wang[5] Lisa Whalley[21], Xuefang Wu[1], Zhijun Wu[6], Pinhua Xie[34], Fumo Yang[35], Qiang Zhang[36], Yanli Zhang[33], Yuanhang Zhang[6], Mei Zheng[6]

[1] School of Geography Earth and Environmental Sciences, University of Birmingham, UK
[2] Institute of Surface Earth System Science, Tianjin University, China
[3] Department of Meteorology, University of Reading, UK
[4] Institut Pierre Simon Laplace, Ecole Polytechnique, France
[5] Institute of Atmospheric Physics, Chinese Academy of Sciences, Beijing, China
[6] College of Environmental Sciences and Engineering, Peking University, Beijing, China
[7] Wolfson Atmospheric Chemistry Laboratories, Department of Chemistry, University of York, York, UK
[8] National Centre for Atmospheric Science, University of York, York, UK
[9] Analytical & Environmental Sciences Division, King's College London, London, UK
[10] Laboratory of Hydrology and Water Management, Ghent University, Coupure Links 653, B-9000 Ghent, Belgium
[11] Lancaster Environment Centre, Lancaster University, Lancaster, UK
[12] School of Earth and Environmental Sciences, The University of Manchester, Manchester, UK
[13] National Centre for Atmospheric Science, The University of Manchester, Manchester, UK
[14] Cambridge Environmental Research Consultants, Cambridge UK
[15] Ricardo Energy & Environment, Harwell, Oxfordshire
[16] Department of Chemistry, University of Cambridge, Cambridge, UK
[17] School of Geosciences, University of Edinburgh, Edinburgh, UK
[18] School of Environment, Tsinghua University, Beijing China
[19] Institute of Chemistry, Chinese Academy of Sciences, Beijing, China
[20] School of International Development, University of East Anglia, Norwich, UK
[21] Department of Chemistry, University of Leeds, Leeds, UK
[22] Centre for Ecology & Hydrology, Penicuik, UK

* Corresponding Author: Zongbo Shi (email: z.shi@bham.ac.uk)
† Also at: Department of Environmental Sciences / Center of Excellence in Environmental Studies, King Abdulaziz University, PO Box 80203, Jeddah, 21589, Saudi Arabia
a Now at: University of Basel, Department of Environmental Sciences, Klingelbergstrasse 27, 4056 Basel, Switzerland
b Now at Jet Propulsion Laboratory, 4800 Oak Grove Drive, Pasadena, CA 91109, USA



[23] School of Earth Sciences, Zhejiang University, Hangzhou, China
[24] Institute of Occupational Medicine (IOM), Edinburgh, UK
[25] School of Earth and Environment, University of Leeds, Leeds, UK
[26] Centre for Cardiovascular Science, Queen's Medical Research Institute, University of Edinburgh,
Edinburgh, UK
[27] School of Environmental Studies, University of East Anglia, Norwich, UK.
[28] Department of Chemistry, University of Leicester, Leicester, UK
[29] Hadley Centre, Met Office, Reading, UK
[30] State Key Laboratory of Coal Resources and Safe Mining & College of Geosciences and
Surveying Engineering, China University of Mining and Technology (Beijing)
[31] School of Public Health, Capital Medical University, Beijing, China
[32] College of Urban and Environmental Sciences, Peking University, Beijing, China
[33] Guangzhou Institute of Geochemistry, Chinese Academy of Sciences, Guangzhou, China
[34] Anhui Institute of Optics and Fine optics, Chinese Academy of Sciences, Hefei, China
[35] Department of Environmental Science and Engineering, College of Architecture and
Environment, Sichun University, Chengdu, China
[36] Department of Earth System Science, Tsinghua University, Beijing, China



## ABSTRACT

APHH-Beijing (Atmospheric Pollution and Human Health in a Chinese Megacity) is an international collaborative project to examine the emissions, processes and health effects of air pollution in Beijing. The four research themes of APHH-China are: (1) sources and emissions of urban atmospheric pollution; (2) processes affecting urban atmospheric pollution; (3) exposure science and impacts on health; and (4) interventions and solutions to reduce health impacts. Themes 1 and 2 are closely integrated and support Theme 3, while Themes 1-3 provide scientific data for Theme 4 on the development of cost-effective solutions. A key activity within APHH-Beijing was the two month-long intensive field campaigns at two sites: (i) central Beijing, and (ii) rural Pinggu. The coordinated campaigns provided observations of the atmospheric chemistry and physics in and around Beijing during November – December 2016 and May- June 2017. The campaigns were complemented by numerical air quality modelling and air quality and meteorology data at the 12 national monitoring stations in Beijing. This introduction paper provides an overview of (i) APHH-Beijing programme, (ii) the measurement and modelling activities performed as part of it in Beijing, and (iii) the air quality and meteorological conditions during the two field campaigns. The winter campaign was characterized by high $PM_{2.5}$ pollution events whereas the summer experienced high ozone pollution events. Air quality was poor during the winter campaign, but less severe than in the same period in 2015 when there were a number of major pollution episodes. $PM_{2.5}$ levels were relatively low during the summer period, matching the cleanest periods over the previous five years. Synoptic scale meteorological analysis suggests that the greater stagnation and weak southerly circulation in November/December 2016 may have contributed to the poor air quality.





## 1.    INTRODUCTION

Air pollution is one of the largest environmental risks. It is estimated that air pollution has led to 7

million premature deaths per year globally (WHO, 2016a, b) and over a million in China (GBD

MAPS Working Group, 2016). Air pollution also has significant impact on the healthcare system

and ecosystems, which cost about 0.3% of global GDP (OECD, 2016). Air pollution related

sickness also reduced productivity and severe hazes lead to closure of transport systems, causing

additional damage to the economy. Total economic losses related to China's $PM_{2.5}$ (particulate

matter with aerodynamic diameter equal to or less than 2.5 µm) pollution in 2007 amounted to 346

billion Yuan (£39 billions, approximately 1.1% of the national GDP) based on the number of

affected Chinese employees whose work time in years was reduced because of mortality, hospital

admissions and outpatient visits (Xia et al., 2016).

Although air pollution in developed megacities sometimes breaks country specific limits and WHO

guidelines, traditional London or Los Angeles type smogs which occurred in the early and mid-20th

centuries are rare in developing cities to the same extent. In the developing countries however, the

rush to industrialisation and rapid growth in vehicle populations have led to serious air pollution

problems that are more complex than the London or Los Angeles smogs. Air pollution is

particularly severe in developing megacities, such as Beijing, where rapid urbanisation has led to a

fast increase in pollution emissions (Guan et al., 2014), on top of regional pollution from industrial

and other anthropogenic activities.

Considerable research effort has led to huge progress in understanding the sources and pollution

processes in megacities in western countries, e.g., major interdisciplinary and multi-institutional

programmes in Mexico City, Paris and London in the last few years (Molina et al., 2010; Beekmann

et al., 2015; Bohnenstengel et al., 2014). Air pollution in megacities in developing countries, in

particular in China have been extensively studied, e.g., in CAREBEIJING (e.g., Liu et al., 2012).



However, our understanding of sources and emissions of key air pollutants such as $PM_{2.5}$ and ozone
plus the interaction of physical and chemical processes in the formation of pollution events in
developing megacities is still far from being accurate or complete.

Beijing's air pollution is different to that in other heavily studied megacities, such as Paris, Mexico
City and London, in a number of ways including the lack of diesel emissions in the inner city, the use
of coal in surrounding rural areas for heating and domestic cooking (Tao et al., 2018), the high
emissions of air pollutants in neighbouring provinces (Hebei and Tianjin) and the high oxidising
power due to the complex chemistry (Zhang et al., 2009; Li et al., 2017; Lu et al., 2018). This makes
Beijing a particularly interesting place to study as it provides a new environment to test our
understanding of urban pollution processes.

Adverse health effect of air pollution is one of the key motivations to control air pollution. Research
has shown that air pollution is one of the leading causes of disease burden in China (GBD MAPS
Working Group, 2016). Especially, particulate pollution, the leading cause of severe air pollution
events in China, has a significant impact on human health and is associated with high mortality
(Zhang et al., 2017a), with considerable proportion of this related to cardiorespiratory diseases
(namely stroke, ischemic heart disease, and chronic obstructive pulmonary disease) (Yang et al., 2013;
Lozano et al., 2013). Despite this increasing evidence base, the adverse health impact of air pollution
remains a complex issue. For instance, the risk assessment of disease burden due to air pollution in
China relied largely on the studies undertaken in Europe and North America, which likely over-
simplifies estimates due to the difference of race, life style, air pollution settings (Lim et al., 2012).
The marked change in air pollution sources and composition between heating and non-heating
seasons, and the differences between urban and rural areas may all lead to different biological
responses in local residents. However, to date, such comparative investigations are largely lacking.





A further limitation of such work is the lack of accurate personal exposure estimates which are crucial
in high quality health studies. This may be especially true when considering household air pollution
(both indoors and outdoors) from traditional biomass and coal stoves which may not be easily
captured by ambient located monitoring instruments (Linn et al., 2001; Brook et al., 2002). To address
current uncertainties and challenges it is essential to improve understanding of the health impact of
air pollution worldwide, and to develop mitigation measures with limited resources on health services.

To address these issues, the UK Natural Environment Research Council (NERC), in partnership with
the National Science Foundation of China (NSFC), UK Medical Research Council (MRC) and UK-
China Innovation Newton Fund funded a major joint research programme – Atmospheric Pollution
and Human Health in a Chinese Megacity (APHH-Beijing). The APHH programme is taking a multi-
disciplinary approach to investigating (1) sources and emissions of urban atmospheric pollution; (2)
processes affecting urban atmospheric pollution; and (3) the exposure and impacts of air pollution on
human health. The scientific understanding from these three themes underpin the development of
interventions and solutions to improve air quality and reduce health impacts.

This special issue "In-depth study of air pollution sources and processes within Beijing and its
surrounding region (APHH-Beijing)" documents the research outcomes of this APHH-Beijing
programme, in particular the atmospheric measurement and modelling aspects. This paper describes
the aims and objectives of APHH-Beijing and presents some of the background air quality and
meteorology observations that form the basis of data interpretation for the whole programme.





## 2.    APHH-BEIJING PROGRAMME OBJECTIVES

The overall aim of APHH-Beijing is to better understand the sources, atmospheric transformations and health impacts of air pollutants in the Beijing megacity and to improve the capability of forecasting air quality and developing cost-effective mitigation measures. Specific objectives include:

- to determine the emission fluxes of key air pollutants and to measure the contributions of different sources, economic sectors and regional transport to air pollution in Beijing

- to improve understanding of the processes by which pollutants are transformed or removed through transport, chemical reactions and photolysis and the rates of formation and conversion of particulate matter via atmospheric reactions

- to improve understanding on how the detailed properties of particulate matter evolve and can influence their physical properties and behaviour in the atmosphere and elucidate the mechanisms whereby those properties may interact and feedback on urban scale and regional meteorology

- to exploit new satellite observations and regional models to place the *in-situ* campaigns into a wider context

- to determine the exposure of Beijing inhabitants to key health related pollutants using personal air pollution monitors and assess the association between air pollution exposure and key cardiopulmonary measures

- to determine the contribution of specific activities, environments and pollution sources to the personal exposure of the Beijing population to air pollutants derived from outdoor sources

- to enhance our understanding of the health effects in susceptible individuals over time periods when there are large fluctuations in pollutants compared with normal controls, and to identify health outcomes of air pollution.

- to estimate economic loss due to both physical and mental impacts of air pollution and examine how Beijing can improve its air quality more cost effectively





### 3.    RESEARCH THEMES AND INTEGRATION WITHIN THE APHH-BEIJING

### PROGRAMME

The APHH-Beijing programme has four themes to address specific objectives (Section 2).

### 3.1    Research Themes

### 3.1.1    Sources and emissions

This topic is addressed by the AIRPOLL-Beijing (Source and Emissions of Air Pollutants in Beijing) project. AIRPOLL aimed to quantify the emission fluxes of key air pollutants in Beijing and the contributions of different sources, economic sectors and regional transport to air pollution in Beijing. Several science topics addressed individual issues, which are integrated to achieve the overall aims. The project carried out two major field measurement campaigns jointly with the AIRPRO (The integrated Study of **AIR** Pollution **PRO**cesses in Beijing) and AIRLESS (Effects of **AIR** pollution on cardiopu**L**monary dis**E**a**S**e in urban and peri-urban re**S**idents in Beijing) projects (section 3.1.2 and 3.1.3) using sites within Beijing (at the Institute of Atmospheric Physics (IAP)) and in the local region (the rural Pinggu site – see 4.1 for site information). During winter and summer sampling campaigns, AIRPOLL measured the concentrations of key tracers and reactive species indicative of sources and chemical pathways at the ground sites. AIRPOLL also analysed the vertical concentration profiles measured in conjunction with data from monitoring sites across Beijing.

As Beijing is subject to long-range transport of pollutants from neighbouring regions, a key aim was to differentiate advected pollutants from local emissions.  Local sources include road traffic, cooking, burning of fossil fuels by industry and for domestic heating.  Secondary pollutants are expected to be largely advected, but the geographic scale of Beijing is sufficient for some formation of secondary pollutants within the city.





During the intensive campaigns, the project measured the fluxes of particulate and gaseous air
pollutants from ground-level sources by sampling on a tower at the IAP site, which are being
compared with estimates taken from the inventory for Beijing. This was complemented by top-down
fluxes inferred from satellite data for nitrogen dioxide, sulphur dioxide and formaldehyde, the latter
indicative of VOC oxidation processes (Palmer et al., 2003; Fu et al., 2007). Through these means,
the emissions inventory are being tested, allowing revisions which are being incorporated into the
atmospheric modelling work.

AIRPOLL also made very detailed on-line and off-line measurements of airborne particles. This
included continuous measurements of size distributions from 1 nm to >10 µm diameter. Large
molecules and molecular clusters were also measured by high resolution mass spectrometry, with a
view to better understanding atmospheric nucleation processes. The project monitored the chemical
composition of particles in real time by Aerosol Mass Spectrometry and analysed the time-integrated
particle samples off-line for major and minor constituents, including organic molecular markers.
AIRPOLL determined the carbon-14 in water soluble organic carbon, water insoluble organic carbon
and elemental carbon in selected time-integrated particle samples with an aim to differentiate fossil
and non-fossil particulate carbon. These data are being brought together for use in receptor modelling
of particulate matter sources, which are compared with other estimates of source contributions to
particulate matter concentrations.

Measured ground-level concentrations and source apportionment are compared with the predictions
of a chemistry-transport model and used to provide a clear distinction between advected regional
pollution and the impact of local sources. Divergences between measured and modelled pollutant
concentrations will be used to provide critical evaluation of emissions inventories, which will be
enhanced iteratively with a view to improving knowledge of the sources and emissions of pollutants
affecting air quality in Beijing. Data from AIRPOLL-Beijing measurement and modelling work will



also contribute to the aims of the AIRPRO project to elucidate the atmospheric physical and chemical
processes determining the measured composition.

**3.1.2    Atmospheric processes**
AIRPRO aims are to study the basic chemical and physical processes controlling gas and aerosol
pollution, localised meteorological dynamics, and the links between them within Beijing's
atmosphere. Once released to air, atmospheric processing controls how pollutants are subsequently
deposited, transformed into secondary pollutants such as $O_3$ and particulate matter (PM) or
transported away from or within the wider Beijing urban area. Previous studies of pollution in Beijing
have shown that it is often perturbation of the physicochemical and dynamic atmospheric conditions
that modulate the most severe air quality events, rather than changes in emissions, for example during
the development of stable inversions or periods of strong photochemistry. Central to the project were
the intensive *in situ* measurements at the IAP meteorological tower (325 m) in Beijing during
November-December 2016 and May-June 2017. We made comprehensive and detailed local
observations of both primary emitted chemicals and particles, radical intermediates and secondary
products, for periods of contrasting local and regional emissions, solar insolation and air temperature.
These data allow the performance of local and regional models of air pollution to be robustly tested,
both for final regulated pollutant outcomes and at a more mechanistic level.

The observations collected with instruments from multiple Chinese and UK research groups included
complementary measurements of key precursor trace gases such as $NO_x$, HONO, $SO_2$, CO, $O_3$, VOCs
and SVOCs, gas phase radicals such as OH, $HO_2$, $RO_2$, and $NO_3$, and PM including chemical (both
on-line and offline analyses), biological, physical and optical properties. Through multiple co-located
surface measurements, there was both instrumental redundancy (e.g. for equipment failures) and
capacity to evaluate through inter-comparison some hard-to-measure atmospheric gases such as OH,
$HO_2$, $N_2O_5$, HCHO and other oxygenated VOCs. The project determined the local *in situ* chemical



processing of air pollution in the contrasting winter/summertime periods alongside overall
atmospheric reactivity, both day and at night, through a combination of modelling and proxy
measurements such as measured ozone production efficiency and OH reactivity.

The IAP tower is critical  as it allowed vertical profiles of key pollutants up to 320 m to be obtained
and,  with  additional remote sensing of composition and meteorology, provided insight into boundary
layer stability and evolution over the diurnal cycle. Quantification of shallow mixed layers proved to
be vital for explaining local surface *in situ* chemical processing and also street level concentrations
of relevance to exposure. The potentially significant vertical gradients anticipated in some chemicals
and PM properties were further quantified using instruments installed on the tall tower and via
profiling gondola measurements. The combined datasets, surface and profiles, provide the basis for
evaluation of model performance, and notably comparisons for those intermediates that provide
indicators of whether secondary pollution production is being correctly simulated.

**3.1.3    Health effects**
This theme is addressed by AIRLESS and APIC-ESTEE (Air Pollution Impacts on Cardiopulmonary
Disease in Beijing: An integrated study of Exposure Science, Toxicogenomics and Environmental
Epidemiology) projects.

AIRLESS aimed to advance air quality and health research in China by bringing together two fields
of research that have made rapid advancements in recent years: measurements of a wide range of
pulmonary and cardiovascular biomarkers in a panel study and personal monitoring of multiple air
pollutants with high spatio-temporal resolution by sensor technology.  AIRLESS is also benefiting
from the use of an extensive range of pollution metrics collected in the Themes 1 and 2 projects.



285 These data are being compared with our personal air quality assessments and be used to further

286 understanding of the nature of the air pollution exposures of residents and how this relates to their

287 health status. The APIC-ESTEE study is examining different aspects of air pollution exposure and

288 health, including population studies and toxicology. One aspect of APIC-ESTEE is investigating the

289 relationship between ambient air pollution and personal exposures, and the impacts of both ambient

290 and personal exposures on subclinical health outcomes. Another part of the study is investigating the

291 real-world exposure-reduction and health impact potential of face-masks, a commonly used personal

292 level intervention seen in Beijing. APIC-ESTEE also carried out laboratory toxicology studies to

293 investigate the toxic mechanisms of PM, and a cohort of mothers and children were recruited to

294 investigate relationships between pre-natal air pollution exposures and birth and infancy outcomes.

295

### 296 3.1.4  Solutions

297 This theme is addressed by INHANCE (Integrated assessment of the emission-health-socioeconomics

298 nexus and air pollution mitigation solutions and interventions in Beijing) project. In recognition of

299 the health and socio-economic issues associated with air pollution, China's State Council authorized

300 a 1.75 trillion Yuan investment package: the Air Pollution Prevention Plan in 2013. INHANCE

301 quantitatively evaluated the performance of China's current air pollution policies wherein the

302 effectiveness of current anti-air pollution measures. INHANCE not only considered physical and

303 mental health impact, direct economic impact, but also the cascading indirect economic losses

304 occurred through inter-industrial and inter-regional linkages on the supply side of the economy.

305 INHANCE established and evaluated interactive relationships among exposure, vulnerability, impact

306 on health, implications for industry and economic consequences.


308 INHANCE compared and qualitatively assessed air quality policies between Beijing and other cities;

309 undertook policy performance assessment modelling; utilised techno-economic inventories for anti-



pollution measures to conduct micro cost-benefit analysis of new policies; measured health and
macroeconomic costs and benefits in mitigating air pollution, and; transformed evidence generated
into practical emission alleviation pathways. On these bases, INHANCE will deliver
recommendations regarding integrated policy design and an assessment for policy cost-effectiveness.

**3.2    Integration Between the Themes**
The APHH-Beijing programme is highly integrated to ensure the biggest possible scientific and
policy impacts. One of the most significant integration activities between the different themes is the
coordinated joint field campaigns at an urban and a rural site in Beijing for Theme 1, 2 and 3 to fully
exploit the complementary measurements and expertise by different research groups, which is
described in the following sections. Theme 1 & 2 are closely related and in many senses inseparable.
For example, our knowledge of the sources and emissions is essential to interpret the processes while
knowledge on the atmospheric physical and chemical processes will help us to more accurately
quantify the source emissions, both via actual flux-based measurements and model evaluation of the
emission inventories. To ensure integration Themes 1 and 2 co-located their rural site at Pinggu as
that was selected for the Theme 3 panel study.

Modelling airborne concentrations of air pollutants within Themes 1 and 2 are fully integrated,
primarily via the UKCA (UK Chemistry and Aerosol), NAQPMS (Nested Air Quality Prediction
Model System) and GEOS-Chem models. Both models simulate spatial and temporal variations of
key air pollutants and will be evaluated using the new observations of pollutant emission fluxes,
updated emission inventories, three-dimensional air quality low cost sensor measurements,
comprehensive composition and physics measurements, as well as new process understandings
generated from the APHH-Beijing programme. Furthermore, Themes 1 and 2 ADMS (Atmospheric
Dispersion Modelling System) modelling results for the campaign periods facilitate estimation of





population exposure in Theme 3. Outcomes of Themes 1, 2 and 3 provide Theme 4 with a more
accurate estimate of pollution costs and help to develop cost-effective air pollution control measures
in Beijing.

The third stream of integration activities involves regular APHH-Beijing programme science and
stakeholder engagement meetings to stimulate collaboration and knowledge transfer between
different themes and stakeholders. Furthermore, sharing of data was made available via a dedicated
depository in Centre for Environmental Data Analysis (www.ceda.ac.uk). All data in the depository
will be made publically available by the end of 2022.

**4.      OVERVIEW OF JOINT FIELD CAMPAIGNS**
The two intensive campaigns were from 5th November to 10th December 2016 and 15th May to 22nd
June 2017. The campaigns were carried out at both urban and rural sites.

**4.1      Site Information**
The winter campaign has two main sites. The urban site (39.97N, 116.38 E) is located in the Tower
Section of Institute of Atmospheric Physics (IAP), Chinese Academy of Sciences; i.e. at the 325 m
meteorological tower. The site, between the fourth and third North ring roads of Beijing (Figure 1),
is a residential area. Typical of central Beijing, there are various roads nearby. To the south, north
and west there are roads about 150 m away. On site there are 2 to 3 floor buildings to the south, east
and west of the tower surrounding by small trees and grasses. There is a canal right to the north of
the site. Further to the west is a park covered mainly by conifer pine trees (Yuan Dynasty Wall
Heritage).



The rural site in Xibaidian village (40.17N, 117.05 E) in north-eastern Beijing, was collocated with
the AIRLESS project cohort. Xibaidian village is about 4 km northwest of Pinggu centre, and about
60 km from IAP. There are many similar small villages nearby. The monitoring station and the clinic
used an unoccupied house at the north end of the village away from significant local combustion
sources. A two-lane road is about 300 m north to the site. With no centralised heating infrastructure
available to the local villages' residents mainly use coal and biomass for heating and cooking in
individual homes.

In the summer, an additional site was operated in Gucheng (39.2N 115.7E), Dingxing County, Hebei
Province. This site, about 120 km to the southwest of central Beijing, is one of the main highly
pollutant transport pathways from Hebei province to Beijing via the southwest passage. The site used
a meteorological observatory in a farm field. The nearest town is about 10 km to the northeast. The
nearest road is 500 m to the north and the nearest village is about 1 km to the west. Several villages
are located around the site.

In addition to the two highly instrumented urban and rural (Pinggu) sites, 21 SNAQ (Sensor Network
for Air Quality) boxes, which measure CO, NO, $NO_2$, $CO_2$, $O_X$, size resolved particulates (0.38-17.4
µm), temperature, relative humidity, wind speed and direction (Popoola et al., 2018), were deployed
during the summer and winter campaigns across the urban and rural areas of Beijing to map air
pollutant variations (red tags, Figure 1). Six additional SNAQ boxes were deployed at six different
heights (8, 32, 102, 160, 260, and 320 m) on the IAP tower from 9-23 November 2016 and 25 January-
31 December 2017.

Figure 1 also shows the location of the 12 national air quality monitoring stations. Hourly data of
criteria air pollutants ($PM_{2.5}$, $PM_{10}$, $SO_2$, $NO_2$, CO and $O_3$) from January 2013 to December 2017





from the stations were also obtained from official sources by Tsinghua University. The closest air
quality station to the urban IAP site is about 3 km away at the Olympic Park.

**4.2 Instrumentation**
**4.2.1 Urban site**
Table 1 lists all instruments deployed during the campaigns at the IAP site. The nine instrument
containers were at ground level on the campus grass. Their locations are shown in Figure 1c. Online
instruments and high volume samplers were deployed at different heights on the meteorological
tower. Most instruments ran during both campaigns. Vertical profiles measurements included HONO
during pollution events using baskets attached to the tower. Additional online measurements and
offline particulate matter samplers were deployed at ground-level, roof of a two storied building to
the west (WB) and in a third-floor laboratory at the south-end of the campus. In addition, high,
medium and low volume samplers were placed on the roof of WB for offline characterization and
source apportionment.

**4.2.2 Rural sites**
At Pinggu, online instruments (Table 3) were run within an air-conditioned room on the ground floor
with inlets on top of the building. High-, medium- and low-volume PM samplers were deployed on a
newly modified flat-roof of the single storey building.
At Gucheng (summer only), a high volume Digitel sampler and a single particle sampler were set up
on a deserted basketball court. An Aethalometer AE33 was located on top of a container at the edge
of the basketball court. CO and $O_3$ were also measured in a nearby container.






**5.    AIR QUALITY DURING THE FIELD CAMPAIGNS**
**5.1    Winter**
During the winter sampling campaign the daily average concentration of $PM_{2.5}$ at IAP using Partisol
gravimetric measurements was 91.2 µg m$^{-3}$ (Table 4) and 94.0 µg m$^{-3}$ from online FDMS (Filter
dynamic measurement system) measurements. The maximum hourly $PM_{2.5}$ concentration was 438
µg m$^{-3}$ (Figure 2). $PM_{2.5}$ concentrations significantly exceeded the both the daily air quality limit of
China (75 µg m$^{-3}$) and WHO (25 µg m$^{-3}$). During the whole winter campaign period, nearly 50% of
the hours had $PM_{2.5}$ mass concentration higher than 75 µg m$^{-3}$ (Figure 2). Online $PM_{10}$ concentration
observed at the Olympic Park national air quality monitoring station was up to 560 µg m$^{-3}$ during the
campaign with an average of 130.6 µg m$^{-3}$. Average concentrations of $NO_2$, $O_3$, $SO_2$ and CO were
$69.7 \pm 33.3$, $16.4 \pm 17.0$ and $14.9 \pm 11.1$ µg m$^{-3}$ and $1.53 \pm 1.02$ mg m$^{-3}$, respectively (Table 4). Most
of the criteria pollutants showed a similar temporal pattern (Figure 2), except $O_3$.

The daily average concentration of $PM_{2.5}$ was 99.7 µg m$^{-3}$ at Pinggu (Table 4; based on Partisol
gravimetric measurement) but as high as 114.0 µg m$^{-3}$ from the BAM measurement. The maximum
hourly $PM_{2.5}$ concentration was 617 µg m$^{-3}$ (Figure 2). Similarly to IAP, nearly 50% of the hours had
$PM_{2.5}$ mass concentrations greater than 75 µg m$^{-3}$. Average concentrations of $NO_2$, $O_3$, $SO_2$ and CO
are $46.4 \pm 25.5$, $22.3 \pm 22.2$, and $15.4 \pm 6.7$ µg m$^{-3}$ and $1.47 \pm 1.17$ mg m$^{-3}$ (Table 4). $PM_{2.5}$ was
slightly higher at the rural site but NO, CO and $SO_2$ were comparable between the two sites. $PM_{2.5}$
and $O_3$ each had similar temporal patterns at the urban and rural sites (Figure 2), indicating a synoptic
scale meteorological impact. The larger difference in the temporal variation of NO, $NO_2$ and $SO_2$ may
reflect the varying contribution of more local sources. Large differences in temporal patterns of air
pollutants were found on 4 December 2016 when $PM_{2.5}$, $SO_2$ and NO concentrations were much
higher at Pinggu than at IAP.



Diurnal cycles of particles, $NO_2$ and CO showed no distinct peak but an increment during the
nighttime, suggesting the possible impact of boundary layer and/or anthropogenic emissions in winter
(Figure 3). The peak NO levels at 7 am are likely caused by the morning rush hour road traffic. $PM_{2.5}$
concentration increased sharply from 6 pm at Pinggu (not shown), suggesting important local
emissions, likely domestic heating and cooking. $SO_2$ and $O_3$ had their highest levels in mid-morning
or at noon (Figure 3).

Variations of particles, $NO_x$ and $SO_2$ show that higher levels of these pollutants when air masses were
from the south or southwest (Figure 4), indicating it was impacted by regional transport. All pollutants,
except $O_3$, had higher mass concentrations when wind speeds were low, suggesting a local source.
The NO wind rose suggests a strong local source with little contribution from long-range transport.
The $O_3$ concentration was higher during northerlies and when the concentrations of other pollutants
such as $NO_x$ and $PM_{2.5}$ were lower (Figure 4).

SNAQ box measurements at six levels (8 to 320 m) during the winter campaign (Figure 5) have
similar overall temporal patterns of CO and NO to that measured by standard gas analyser (Figure 2).
In most cases, the air pollutant levels are similar at different levels of the tower. There are notable
differences in NO, CO and $CO_2$ on 11, 12 and 16 / 17 November, which suggests that the mixed layer
height was low (e.g., <150 m). Interestingly, the $O_x$ ($NO_2 + O_3$) levels are relatively homogeneous
across the different levels. These measurements have implications on the role atmospheric chemistry
play in transformation of species in the boundary layer, and the measurements also provide useful
information that confirm mixed layer height determinations from independent methods such as the
ceilometer.



According to the meteorological standards (QX/T113-2010), haze is defined as: i) visibility < 10 km
at relative humidity (RH) <80%; or ii) if RH is between 80 and 95%, visibility < 10 km and $PM_{2.5}$ >
75 µg m$^{-3}$. During the winter campaign  640 of the 1633 h were classified as haze using visibility data
from Beijing Capital Airport (Figure 6); within the haze hours 75% had $PM_{2.5}$ greater than 75 µg m$^{-}$
$^3$ (Area A, Figure 6) and the rest had a visibility less than 10 km but with a RH <80% (Area B, Figure

465   6).


Characteristics of five major haze events during the winter campaign (Figure 2) include that $PM_{2.5}$,
$NO_2$, $SO_2$ and CO had similar trends but  $O_3$ levels dropped to very low concentration (<2 ppb). The
events are defined in Table 2.

**5.2      Summer**
Concentrations of air pollutants excluding ozone during the summer campaign were much lower than
in  winter (Figure 7, Table 4).  Average daily concentration of $PM_{2.5}$ and $PM_{10}$ at IAP were 31.4 ±
14.7 and 74.9 ± 29.3 µg m$^{-3}$ (based on gravimetric method), respectively. These levels were slightly
higher than at Pinggu (27.8 ± 13.3 and 62.9 ± 29.3 µg m$^{-3}$). Concentrations of ozone were four to five
times higher during the summer campaigns (106.9 ± 71.6 µg m$^{-33}$ at IAP, and 91.8 ± 62.7 µg m$^{-33}$ at
Pinggu) than in the winter campaign.  Average concentration of $NO_2$, $SO_2$ and CO are 41.3 ± 23.5
and 6.3 ± 6.8 µg m$^{-3}$and 0.61 ± 0.32 ,g m$^{-3}$at IAP (Table 4). The concentration of $NO_2$ and CO were
lower at Pinggu while that of $SO_2$ was similar. Most of the criteria pollutants showed a similar
temporal pattern (Figure 2), except $O_3$.

Diurnal patterns of NO, $NO_2$, and CO at IAP showed a distinct peak in the early morning, suggesting
the contribution of traffic emissions (Figure 7). $O_3$ and $O_x$ concentration peaked in mid-afternoon.



The IAP PM$_{2.5}$ wind rose suggests both local and regional sources (from the south and south-east
direction) impact the site (Figure 4). Unlike winter, high ozone concentrations occur during
southerlies to southwesterlies, suggesting a regional source of this pollutant. NO and NO$_x$ were
largely from local sources during the summer campaign.
Characteristics of two minor haze events (IAP) during the summer campaign (Figure 7) are shown in
Table 2.

**5.3        Air quality in Wider Beijing Megacity During the Field Campaigns**
Average concentrations of air pollutants (PM$_{2.5}$, PM$_{10}$, NO$_2$, CO, SO$_2$ and O$_3$) at IAP and Pinggu
during the two field campaigns were similar to long term averages for these times of year at the 12
national air quality monitoring sites for 2013-2017 (Table 4).

To assess if the IAP air quality is broadly representative of the wider Beijing megacity, variables are
correlated with the 12 national air quality station data (Figure 8). A high correlation occurs with PM$_{2.5}$
across all sites except the rural background air quality station at Ming Tombs; PM$_{10}$, CO and NO$_2$ at
the urban sites are highly correlated but not with the rural and suburban sites suggesting a more local
source for these pollutants, comparing to PM$_{2.5}$ and O$_3$; SO$_2$ between sites have lower correlation
comparing to all other pollutants. The particularly high correlation of PM$_{2.5}$ and O$_3$ across almost all
sites indicates a regional pollution phenomenon for the two pollutants. These results suggested that
the air quality at the IAP urban site was broadly consistent with those at the other urban sites.

In general, PM$_{2.5}$ mass concentrations are similar at all the urban sites including IAP but higher than
at the suburban and rural background national monitoring site (Ming Tombs, G2) (Figure 9). The
Pinggu rural site in this study, has high PM$_{2.5}$ pollution in the winter campaign but has the lowest
concentrations during the summer campaign. This suggests that local anthropogenic sources have a



major impact on $PM_{2.5}$ at this site during the winter campaigns. Source apportionment results, notably
high time resolution data are being used to explore this.

The closest national air quality station (Olympic Park, or Aotizhongxin in Chinese Pingyin) to IAP
has highly correlated $PM_{2.5}$ concentration. This suggests that national air quality stations are of
sufficient quality to provide valuable information on the spatial and temporal variation of key
pollutants to supplement campaign measurements.

Table 4 show the IAP concentrations data for all air quality variables are very close to the 12 national
air quality monitoring stations mean. This lends further confidence that the chosen urban site
represented well the overall pollution in the Beijing megacity.

**6.      SYNOPTIC SCALE METEOROLOGY DURING THE FIELD CAMPAIGNS**
Given the importance of horizontal advection and wet deposition to air quality in Beijing, the synoptic
circulation patterns are clearly important (Miao et al., 2017; Wu et al., 2017; Zhang et al., 2012). To
provide the synoptic context of the APHH-China observations, the daily mesoscale flow patterns are
classified (Section 6.1) and put into context using a 30-year climatology (Section 6.2).

**6.1    Synoptic Circulation Types**
Circulation types (CT) are classified using the classification software by the COST Action 733
"Harmonisation and Applications of Weather Type Classifications for European regions" (Philipp et
al., 2010) with (ECMWF Re-Analysis) ERA-Interim 6-h 925 hPa geopotential reanalysis data (Dee
et al., 2011) at its native 0.75° spatial resolution for the domain of interest (103-129° E, 31 - 49° N)
centred on Beijing (40° N, 116.5° E) covering the period 1988-2017. ERA-Interim 10 m U and V



wind components are used to facilitate interpretation of the flow patterns. Of the COST733 methods
(Huth et al., 2008; Philipp et al., 2010, 2016; Tveito and Huth, 2016) two are used: T-Mode PCA
(Principal Component Analysis) and SANDRA (Simulated Annealing And Diversified
RAndomization clustering). The former have been used in Beijing previously (e.g. Miao et al., 2017;
Zhang et al., 2012). The latter is considered to perform well in clustering pressure fields and
discriminating environmental variables (e.g. Demuzere et al., 2011; Philipp et al., 2016).
Classification is performed with the number of CTs ranging from 7 to 18. 11 CTs from the SANDRA
method are selected (Figure 11; Table 5) to adequately represent the general flow conditions around
Beijing during the 30 y climatology period (Beck and Philipp, 2010). The CTs are re-ordered
according to the daily median $PM_{2.5}$ concentration observed at the Olympic Park (i.e. Aotizhongxin)
(Figures 1 and 12) in 2013-2017 with the predominant CTs estimated from midday-midday, i.e. with
a 12 h time lag.

As expected, the CTs that occurred during the two field campaign periods are different (Figures 12
and 13). During the winter field campaign, most frequent circulation type was CT 10 (25 % of the 6
h periods) and often preceded by a period of CT 11 (total 16%). Circulation types 9-11 are associated
with air masses that may stagnate over the Beijing urban area (Figure 11). However, CT 9 did not
occur in winter (or the summer) field campaign. CT 1 accounted for 16% of the time, with CT 2 (1
%) are associated with the Asian winter monsoon which brings cold and dry air masses to eastern
China. North-westerly flow over Beijing is driven by high pressure in the west of the domain (Figure
11). After these CT 3, 4, 6, and 5 were the most frequent in the winter campaign (12.5, 11.8, 8.3 and
7.6 % of the time, respectively). CTs 3 and 5 are associated with relatively low pressure in the
northeast (Sep-May period). CTs 4 and 6 have a further reduction in atmospheric pressure in the NE.
The remaining 6 h period was classified as CT 7, which occurs when winds are oriented westward
from the Bohai Sea.





During the summer campaign (Figure 12b), the most frequent CT were 5, 8, 6, 7 (34, 32, 12, 11 % of
the time, respectively). CT 8, which did not occur during the winter campaign period, is like CT 6
associated with the summer monsoon advecting moist warm air from the South and Southeast (Figure
11). The other two were CT 1 and 4 (7 and 4 %, respectively).  During spring and summer (Mar-
Aug,) CT 4 winds start to turn over the Yellow Sea, weakening the NW flow over Beijing.

In comparison to the field campaigns, during the period 1988-2017 the CT frequencies range from
7.2% (CT 2, 10) to 12.9% (CT 8) with clear seasonal variations in their occurrence (Figure 13).

**6.2    Synoptic circulation and Air Quality**
The 11 CTs (Section 6.1) are clearly associated with distinct air quality conditions based on analysis
of hourly air quality data for 2013-2017 at one of the national urban air quality station (G4, Olympic
Park, Figures 1 and 12). Relatively lower $PM_{2.5}$ concentrations occur (Figure 13b) under NE flow
conditions (CTs 1-5), and higher concentrations during southerly flow (CTs 6-8, 10). The highest
$PM_{2.5}$ concentrations occur during the heating season associate with stagnation (CT 9, 11). Ozone
levels are highest during CTs 5-8 (Figure 13c) as these predominate during spring and summer
(Figure 13d).

Similarly, the average mixed layer height observed at IAP (Table 1) varies with season and CT type
(Figure 13a). In the Oct 2016 – Sept 2017 period (Figure 13e), the relative frequency of CTs differs
slightly from the long-term climatology (Figure 13d). In December 2016, clear air advection from the
NE (CTs 1-3) was less frequent than in the 30-y climatology. However, stagnation with a weak
southerly component (CTs 9 and 11) was more frequent (Figure 13f), thus favouring haze with a large
positive (40%) $PM_{2.5}$ anomaly (Figure 14g, cf. 5 y average, 2013-2017). In June 2017, south-north
contrasts in geopotential were apparently reduced so CT 6 was 24% less frequent, while CTs 4, 7,





and 8 were more frequent. This had minimal effect of $PM_{2.5}$; the slight increase in $O_3$ (by 9.5%, Figure
13g) might be explained by associated cloud cover differences.

**6.3      Meteorological Conditions During the Field Campaigns**
To assess how local-scale flow related to ERA-Interim fields (section 6.1), the link between the coarse
gridded data and tower-based sonic anemometer observations is explored based on wind roses (Figure
14). The 30 y climatology (Figure 13a, d) confirms the clear seasonality in wind direction affecting
the occurrence of CTs discussed (Sect. 0), i.e. during winter intensive campaign period (5 November
– 10 December) north-easterly flow clearly dominates while southerly wind directions are most
common during the summer campaign period (15 May – 22 June). The wind roses for winter 2016
and summer 2017 (Figure 14b, e) are slightly nosier, however, indicating similar tendencies as the
climatology. The general large-scale patterns are consistent with the in-situ wind measurements
(Figure 14c, f). However, a slight diversion towards northerly and south-westerly flow and lower
wind speeds occurred in winter and summer (Figures 14c and f), respectively, when compared to the
larger scale data (Figures 14b and d).  In addition, south-westerly flows were more frequent in winter
2016 (Figures 14b and c) than the 30 year average climatology (Figure 14a), which had the potential
to bring more polluted air in the upwind Hebei province to the observation sites in Beijing.

At 102 m, the flow is consistent with northerlies and north-westerlies in the winter campaign and
dominantly southerly and easterlies during the summer campaign (Figure 15). The measured hourly
mean wind speed, temperature and relative humidity were 3.1 m s$^{-1}$, 8.3 ºC and 43.8 % in winter, and
3.6 m s, 25 ºC and 46.7 % in summer, respectively. Typical diurnal patterns were observed with
higher wind speed and temperature during the day and RH at night. During the winter haze events the
120 m wind speed were low (an average of 1.8 m s$^{-1}$) and mainly from the south-west direction
(Figures 15 and 2).



**6.4    Pollution Climatology of the Campaign Periods**
To determine how representative the campaign periods were of the selected seasons in Beijing,
pollutant levels were compared with those from the same period each year over the 2013-2017 period.
The NAQPMS model was run for the full 5-year period driven by NCEP meteorology and using
temporally varying emissions for a single year that is broadly representative of 2017 conditions. Use
of annually invariant emissions permits the effect of differing meteorology on pollutant levels to be
assessed. The frequency distribution of $PM_{2.5}$ for each campaign period for each year is shown in
Figure 16.  $PM_{2.5}$ in winter 2016 is very similar in characteristics to that in 2014, and both years show
50% greater PM levels than in 2013 or 2017. However, pollutant levels are substantially lower than
in the same period in 2015, when three extended pollution episodes led to period-mean $PM_{2.5}$ that
was almost twice as large. In contrast, the summer period in 2017 was relatively clean, with $PM_{2.5}$
levels very similar to 2015, and about 25% less than in 2013, 2014 or 2016.

**Data depository:**
**http://catalogue.ceda.ac.uk/uuid/7ed9d8a288814b8b85433b0d3fec0300**

**ACKNOWLEDGEMENT**
Funding is provided by UK Natural Environment Research Council, Medical Research Council and
Natural Science Foundation of China under the framework of Newton Innovation Fund
(NE/N007190/1 (R Harrison, Z Shi, W Bloss); NE/N007077/1 (W Bloss)); NE/N00700X/1 (S
Grimmond), NE/N007018/1 (F Kelly); NSFC Grant 81571130100(T Zhu), NE/N007115/1 (A C
Lewis, A R Rickard, D C Carslaw); NE/N006917/1 (J D Lee, J F Hamilton, R E Dunmore);
NE/N007123/1 (J Allan, C Percival, G McFiggans, H Coe); NE/N00695X/1 (C Percival, H Coe, G
McFiggans, J Allan); NE/N006976/1 (N Hewitt, O Wild); NE/N006925/1 (O Wild); NE/N006895/1
(D Heard, L Whalley); NE/N00714X/1 (D Guan), NE/N007182/1 (M Loh); and NE/N006879/1 (P
Palmer). Other Grant supports from Newton Fund/Met Office CSSP-China (S Grimmond; R Doherty





and Z Shi), Royal Society Challenge Grant (CHG/R1/17003, Palmer) and NERC (NE/R005281/1,
Shi) are acknowledged. Field help from Kjell zum Berge, Ting Sun at Reading University are also
acknowledged. Other staff and students at all involving institutions are acknowledged for their
contribution to the field campaigns and programme.

**AUTHOR CONTRIBUTIONS**

ZS drafted the manuscript and is the science coordinator of the APHH-Beijing programme. RMH,
KBH, ACL, PQF, TZ, FJK, ML, ZWS, DBG and ST are lead PIs of the five research projects who
led the funding applications and the research. They also drafted section 2. TV plotted many of graphs
and carried out the data analysis. SK, SG and MD carried out analysis and wrote section 6.1-6.2; and
YLW and OW carried out modelling and plotted Figure 16. PFQ, JL and ZT led the air quality
measurements at the two measurements sites. SY, JL, RED, LR, DL, JA, DB, WJ, LC, LC, HC, TD,
FKD, BZG, JFH, MH, DH, CNH, MH, DSJ, XJJ, RJ, MK, LK, BL, LC, JL, WJL, KDL, GM, MM,
GM (Mills), EN, BO, CP, PIP, OP, CR, LYS, YS, SRT, QQW, WHQ, XMW, ZFW, LW, XFW,
ZJW, PHX, FMY, QZ, YLZ and MZ contribute to the field observations, laboratory measurements
and / or modelling. ZS, SG, RMH., ZT, JL, OW, JA, JB, WJB, DC, DCC, HC, TD, RD, FKD, PQF,
MFG, DBG, JFH, KBH, MH, DH, CNH, MH, XJJ, RJ, MK, FJK, LK, ACL, JL, ML, KL, GM
(Mann), GM (McFiggans), MM, PM, EN, FO, PIP, CP, CR, ARR, LYS, GYS, DS (Spracklen), DS
(Stevenson), YS, ZWS, ST, SRT, XMW, ZFW, LW, ZJW, PHX, QZ, YHZ and MZ contributed to
the funding applications, programme meetings and relevant programme research and/or supervision.




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

Contributions of trans-boundary transport to summertime air quality in Beijing, China, Atmos.
Chem. Phys., 17, 2035-2051, 2017.
Xia, Y., Guan, D., Jiang, X., Peng, L., Schroeder, H., Zhan, Q.:  Assessment of socioeconomic costs
to China's air pollution. Atmos. Environ. 139, 147-156, 2016.
Xie, C., Xu, W., Wang, J., Wang, Q., Liu, D., Tang, G., Chen, P., Du, W., Zhao, J., Zhang, Y.,
Zhou, W., Han, T., Bian, Q., Li, J., Fu, P., Wang, Z., Ge, X., Allan, J., Coe, H., and Sun, Y.:
Vertical characterization of aerosol optical properties and brown carbon in winter in urban Beijing,
China, Atmospheric Chemistry and Physics Discussions, 1-28, 10.5194/acp-2018-788, 2018.
Yang, G., Wang, Y., Zeng, Y., Gao, G. F., Liang, X., Zhou, M., Wan, X., Yu, S., Jiang, Y.,
Naghavi, M., Vos, T., Wang, H., Lopez, A. D., Murray, C. J. L.:  Rapid health transition in China,
1990-2010: findings from the Global Burden of Disease Study 2010, The Lancet, 381, 1987-2015,
1063 2013.
Yu, J., Yan, C., Liu, Y., Li, X., Zhou, T., Zheng, M.:  Potassium: A Tracer for Biomass Burning in
Beijing? Aerosol Air Qual. Res., 18, 2447-2459, 2018.



Yue, S., Ren, H., Fan, S., Sun, Y., Wang, Z. and Fu P.:  Springtime precipitation effects on the
abundance of fluorescent biological aerosol particles and HULIS in Beijing, Sci. Rep., 6, 29618,
10.1038/srep29618, 2016.
Zhang, Q., Streets, D. G., Carmichael, G. R., He, K. B., Huo, H., Kannari, A., Klimont, Z., Park, I.
S., Reddy, S., Fu, J. S., Chen, D., Duan, L., Lei, Y., Wang, L. T., and Yao, Z. L.:  Asian emissions
in 2006 for the NASA INTEX-B mission, Atmos. Chem. Phys., 9, 5131-5153, 2009.
Zhang, J. B., Xu, Z., Yang, G., and Wang, B.: Peroxyacetyl nitrate (PAN) and peroxypropionyl
nitrate (PPN) in urban and suburban atmospheres of Beijing, China, Atmos. Chem. Phys. Discuss.,
1077    11, 8173-8206, 2011.
Zhang, J. P., Zhu, T., Zhang, Q. H., Li, C. C., Shu, H. L., Ying, Y., Dai, Z. P., Wang, X., Liu, X. Y.,
Liang, A. M., Shen, H. X. and Yi, B. Q.: The impact of circulation patterns on regional transport
pathways and air quality over Beijing and its surroundings, Atmos. Chem. Phys., 12, 5031-5053,
1082    2012.
Zhang, Q., Jiang, X., Tong, D., Davis, S. J., Zhao, H., Geng, G., Feng, T., Zheng, B., Lu, Z.,
Streets, D. G., Ni, R., Brauer, M., van Donkelaar, A., Martin, R. V., Huo, H., Liu, Z., Pan, D., Kan,
H., Yan, Y., Lin, J., He, K.and Guan, D.:  Transboundary health impacts of transported global air
pollution and international trade, Nature 543, 705-709, 2017a.
Zhang, Y., Ren, H., Sun, Y., Cao, F., Chang, Y., Liu, S., Lee, X., Agrios, K., Kawamura, K., Liu,
D., Ren, L., Du, W., Wang, Z., Prevot, A. S. H., Szidat, S., and Fu, P.: High contribution of non-
fossil sources to sub-micron organic aerosols in Beijing, China, Environ. Sci. Technol., 2017b.
Zhao, W., Kawamura, K., Yue, S., Wei, L., Ren, H., Yan, Y., Kang, M., Li, L., Ren, L., Lai, S., Li,
J., Sun, Y., Wang, Z., and Fu P.:  Molecular distribution and compound-specific stable carbon
isotopic composition of dicarboxylic acids, oxocarboxylic acids and α-dicarbonyls in PM$_{2.5}$ from
Beijing, China, Atmos. Chem. Phys., 18, 2749–2767, 2018.
Zhou, W., Zhao, J., Ouyang, B., Mehra, A., Xu, W., Wang, Y., Bannan, T. J., Worrall, S. D.,
Priestley, M., Bacak, A., Chen, Q., Xie, C., Wang, Q., Wang, J., Du, W., Zhang, Y., Ge, X., Ye, P.,
Lee, J. D., Fu, P., Wang, Z., Worsnop, D., Jones, R., Percival, C. J., Coe, H., and Sun, Y.:
Production of N$_2$O$_5$ and ClNO$_2$ in summer in urban Beijing, China, Atmos. Chem. Phys., 18, 2018.



**TABLE LEGENDS:**

**Table 1:** Overview of measurements in APHH-Beijing at the urban site.

**Table 2:** Haze periods during the summer and winter campaign periods.

**Table 3:** Overview of measurements at the Pinggu site.

**Table 4:** Average air quality variables at IAP, Pinggu and 12 national monitoring sites (12N) during the field campaigns (10 November – 11 December 2016; and 21 May – 22 Jun 2017). The 12 national sites five-year mean concentrations for same times of the years (12N -5Y) and for the same time of the year (campaign period) (12N-campaign). Data are mean ± s.d. (range).

**Table 5:** Mean and standard deviation (sd) of climatological conditions in Beijing for each circulation type (CT) for 1988-20 17 from Era Interim data with frequency of the CT during the W (winter) and S (summer) campaigns (% of 6 h periods (p)) compared to A- long- term 1988-2017.

**FIGURE LEGENDS**

**Figure 1:** Study area topography (source: googlemap) of Beijing / Tianjing / Hebei region (a) with the rectangle showing enlarged study area; locations of measurement sites (Institute of Atmospheric Physics (IAP)– urban Beijing, Pinggu – rural Beijing; and Gucheng – upwind site in Hebei province), SNAQ box sites (red symbols) and the 12 national air quality monitoring stations (G1 to G12, blue symbols) (b); locations of the 9 containers at IAP (c) – instrumentation at each container is shown in Table 1. The shaded area shows the Beijing buildup area. (Source: a and b - Goggle Map topographic background imagery; c – taken by Siyao Yue from IAP).

G1: Wangshouxigong; G2: Dingling; G3: Dongsi; G4: Tiantan; G5: Nongzhanguan; G6: Guanyuan; G7: Haidianquwanliu; G8: Shunyxicheng; G9: Huairouzhen; G10:Changpingzhen; G11: Aotizhongxin (Olympic Park); G12: Gucheng. Categories: Urban: G1, G3, G4, G5, G6, G7, G8, G11, G12; Suburban: G9, G10; Rural: G2.

**Figure 2:** Time-series of air quality variables at the urban and rural sites during the winter campaign; Five haze events are indicated (shading).

**Figure 3:** Diurnal patterns of gaseous pollutants normalized by average concentrations at IAP during winter and summer campaigns. Line shows the mean concentrations and shaded area as 95% confidence interval in the difference in mean concentrations

**Figure 4:** Air pollutants concentration (colour) with wind direction (angle) and wind speed (m s$^{-1}$) at IAP during the winter and summer campaigns. Data are hourly in time resolution and were from 10 November to 11 December 2016 (winter) and 21 May to 22 June 2017 (summer). The colour scale is for "weighted.mean" where the mean wind speed/direction bin is multiplied by the bin frequency and divided by total frequency.



| 1155 | **Figure 5:** | Time series of $CO_2$, CO, NO, $O_x$ ($NO_2+O_3$) and wind speed at six heights (colour) measured with SNAQ boxes on the IAP tower during the winter intensive field campaign. |
| 1156 | | |
| 1157 | | |
| 1158 | | |
| 1159 | **Figure 6:** | Hourly $PM_{2.5}$ mass concentrations versus visibility (at the Beijing Capital Airport) during the winter campaign. Data source: visibility downloaded using R-"worldmet" package: date of last access: 27/02/2018). |
| 1160 | | |
| 1161 | | |
| 1162 | | |
| 1163 | **Figure 7:** | Time-series of air quality variables at the urban and rural sites during the summer campaign. Two minor haze events are indicated (shading). |
| 1164 | | |
| 1165 | | |
| 1166 | **Figure 8:** | Correlations between the air quality at IAP, PQ and 12 monitoring station around Beijing. Stations G1-G12 (Figure 2) are labelled 01-12, PG = Pinggu. |
| 1167 | | |
| 1168 | | |
| 1169 | **Figure 9:** | Spatial distribution of hourly mean concentration of $PM_{2.5}$ in Beijing during two sampling campaigns. |
| 1170 | | |
| 1171 | | |
| 1172 | **Figure 10:** | Hourly $PM_{2.5}$ at IAP (roof of a two storied building) and the neighbouring Olympic park national air quality monitoring station during the winter and summer intensive field campaigns. |
| 1173 | | |
| 1174 | | |
| 1175 | | |
| 1176 | **Figure 11:** | ERA-Interim (1988-2017) average 925 hPa geopotential with 10 m horizontal wind vector for 11 circulation types classified for Beijing (municipal boundary thin solid line) surroundings (103-129° E, 31 - 49° N) determined with the SANDRA method (COST733 class software). Frequency of occurrence is given in cluster caption. For discussion of conditions associated with each CT see section 6.1. |
| 1177 | | |
| 1178 | | |
| 1179 | | |
| 1180 | | |
| 1181 | | |
| 1182 | **Figure 12:** | Time series of circulation types (CTs) during the two field campaigns: (a) winter and (b) summer. The 11 CTs are shown in Figure 11. See text for more description. |
| 1183 | | |
| 1184 | | |
| 1185 | **Figure 13:** | Analysis by circulation type (CT; Sect. 0) of: (a) daily maximum mixed layer height (MLH) determined from ALC observations at IAP between November 2016 – June 2017 (analysis method, Kotthaus and Grimmond, 2018b); concentration of (b) $PM_{2.5}$ and (c) $O_3$ at at the Olympic Park (i.e. Aotizhongxin) in 2013-2017 from the national air quality network; occurrence of CTs in (d) 1988-2017 and (e) Oct 2016 – Sept 2017; (f) anomaly of CT frequency during Oct 2016 – Sept 2017 compared to 30 y climatology; and (g) anomaly of $PM_{2.5}$ and $O_3$ during Oct 2016 – Sept 2017 compared to 5 y (2013-2017) average (same data as in b, c). |
| 1186 | | |
| 1187 | | |
| 1188 | | |
| 1189 | | |
| 1190 | | |
| 1191 | | |
| 1192 | | |
| 1193 | | |
| 1194 | **Figure 14:** | Beijing wind roses: (a, b, d, e) ERA-Interim 10 m horizontal wind (40° N, 116.5° E) and (c, f) sonic anemometer (Table 1) at IAP 320 m agl for (a) 5 November – 10 December in 1988-2017, (d) 15 May – 22 June in 1988-2017, (b, c) 5 November – 10 December 2016, and (e, f) 15 May – 22 June 2017. |
| 1195 | | |
| 1196 | | |
| 1197 | | |
| 1198 | | |
| 1199 | **Figure 15:** | Hourly meteorological variables measured at 120 m during the (a) winter and (b) summer campaigns. The shaded areas highlighted the haze periods (Table 3, Figures 2 and 7). |
| 1200 | | |
| 1201 | | |
| 1202 | | |
| 1203 | **Figure 16:** | Frequency distribution of $PM_{2.5}$ in Beijing over the winter (left) and summer (right) campaign periods from the NAQPMS model compared with those from the same periods over the past five years under the same emission conditions. |
| 1204 | | |
| 1205 | | |
| 1206 | | |





**Table 1:** Overview of measurements in APHH-Beijing at the urban site.

| | Instrument | Measurements | Institute | References |
|---|---|---|---|---|
| *Container 2* | FAGE | OH (Chem and Wave)[X], $HO_2$, $RO_2$ | Leeds | Whalley et al. (2010) |
| | OH reactivity | OH reactivity | Leeds | Stone et al. (2016) |
| | Spectral radiometer | Photolysis rates | Leeds | Bohn et al. (2016) |
| | Filter radiometer | $J(O^1D)$ | Leeds | Bohn et al. (2016) |
| | Dew point hygrometer | Water vapour | Leeds | Whalley et al. (2010) |
| | Davis met station | Wind speed, direction, temp, RH, pressure | Leeds | |
| | Vaisala CL31 ALC Ceilometer [+] | Cloud-base height, mixing height, attenuated backscatter profiles | Reading | Kotthaus and Grimmond (2018a) |
| | Personal air monitors (PAMS) | CO, NO, $NO_2$, $PM_1$, $PM_{10}$, $PM_{2.5}$ | Cambridge | Moore et al. (2016) |
| | MicroPEMs | Personal PM exposure | IOM | Sloan et al. 2015 |
| *Container 2* | DC-GC-FID | C2-C7 VOCs and oVOCs | York | Hopkins et al. (2011) |
| | GCxGC FID | C6 - C13 VOCs and oVOCs | York | Dunmore et al. (2015) |
| | TEI 42i | NO | Birmingham | |
| | Teledyne CAPS | $NO_2$ | York | |
| | TEI 42c | Total $NO_y$ | York | |
| | TEI 49i | $O_3$ | York | |
| | TEI 43i | $SO_2$ | York | |
| | Sensor box | CO | York | Smith et al. (2017) |
| | BBCEAS | HONO, $NO_3$, $N_2O_5$ | Cambridge | Le Breton et al. (2014) |
| *Container 3* | LOPAP | HONO | Birmingham | Crilley et al. (2016) |
| | LIF HCHO | HCHO | Leeds | Cryer et al. 2016 |
| | LOPAP | HONO | IC-CAS | Zhang et al. (2018) |
| | GC-MS | Organic nitrates | East Anglia | Mills et al. (2016) |
| | ROS online analyser | Reactive Oxygen Species | Cambridge | Wragg et al. (2016) |
| *Container 4 [*]* | FAGE | OH (wave)[x], $HO_2$ | Peking | Lu et al., 2012 |
| | FAGE | OH (chem)[x] | Peking | Tan et al., 2017 |
| | TEI 42i | NO | Peking | Tan et al., 2017 |
| | Teledyne CAPS | NO2 | Peking | |
| | TEI 42c with Moly converter | $NO_2$ | Peking | |
| | TEI 49i | $O_3$ | Peking | |
| | TEI | CO | Peking | |
| | Spectral radiometer | Photolysis rates | Peking | |
| | GC-ECD | PAN | Peking | Zhang et al., 2011 |
| | GC-MS | VOCs | Peking | Wang et al., 2015a |



| | Instrument | Measurement | Institution | Reference |
|---|---|---|---|---|
| **Container 5 *** | H-TDMA/V-TDMA | Hygroscopicity/volatility | Peking | Wu et al., 2013 |
| | SMPS+APS | Particle Number size distribution | Peking | Wu et al., 2016 |
| | Particle size magnifier | Size distribution of < 3nm particles | Peking | Vanhanen et al., 2011 |
| | IGAC-IC | Water-soluble ions | Peking | Yu et al. (2018) |
| | Xact | Metal | Peking | Yu et al. (2018) |
| | Sunset OC/EC | EC/OC | Peking | Zhang et al. (2017b) |
| **Container 6** | IBBCEAS | HONO, $NO_2$ | AIOFM | Duan et al. (2018) |
| | CRDS | $NO_3$ and $N_2O_5$ | AIOFM | Li et al. (2018) |
| | Nitrate Api-TOF-CIMS | Organics, clusters (HOMs) | Birmingham | Junninen et al. (2010) |
| | SMPS | Particle size distribution | Birmingham | Shi et al. (1999) |
| | Particle size magnifier | Size distribution of < 3 nm particles | Birmingham | Vanhanen et al. (2011) |
| **Container 7** | Fast $NO_x$ | $NO_x$ fluxes | York | Vaughan et al. (2016) |
| | AL5002 CO analyser | CO fluxes | York | Gerbig et al. (1999) |
| | HR-TOF-AMS | Fluxes of $PM_1$ non-refractory (NR) species | CEH | Nemitz et al. (2008) |
| | SP2 | BC fluxes | Manchester | Liu et al. (2017) |
| | PTR-TOF-MS | VOC fluxes | GIG Lancaster | Huang et al. (2016) |
| | SYFT-MS Voice 200 Ultra | VOC fluxes | York | Storer et al. (2014) |
| **Container 8** | SMPS3968-APS3321 | Particle number size distribution | BNU | Du et al. (2017) |
| | H/V TDMA | Particle hygroscopicity | BNU | Wang et al. (2017b) |
| | CCNC-100 | CCN | BNU | Wang et al. (2017b) |
| | PAX (870nm) | Extinction & absorption coefficient | IAP | Xie et al. (2018) |
| | Ammonia analyzer | $NH_3$ | IAP | Meng et al. (2018) |
| | Sunset OC/EC analyzer | Online OC/EC | IAP | Zhang et al. (2017b) |
| **Container 9** | Iodide FIGAERO-TOF-CIMS | Particle and gas phase molar molecule | Manchester | Le Breton et al. (2018) |
| | CPMA-SP2 | Black carbon mass and mixing state | Manchester | Liu et al. (2017) |
| | Micro reactor | oVOCs | York | Pang et al. (2014) |
| **Tower ~100 m** | QCL $NH_3$ | Ammonia fluxes | CEH | McManus et al. (2010) |
| | IRGA LiCOR-7500 | $CO_2$ / $H_2O$ flux | CEH | McDermitt et al. (2011) |
| | DMT UHSAS | Size resolved particle flux (0.06-1 µm) | CEH | Deventer et al. (2015) |





| | Instrument | Measurement | Institution | Reference |
|---|---|---|---|---|
| | TSI APS3021 | Size-resolved particle flux (0.5-25 µm) | CEH | Nemitz et al. (2002) |
| | TSI CPC3785 | Total particle number flux | CEH | Petäjä et al. (2006) |
| | ROFI | O$_3$ flux | CEH | Coyle et al. (2009) |
| | Sonic anemometer R3-50 | Turbulence, sensible heat flux | CEH | Högström and Smedman (2004) |
| | WXT530 weather station | T, P, RH, wind speed & direction, precipitation | CEH | |
| | 2B O$_3$ analyser | O$_3$ concentration | CEH | Johnson et al. (2014) |
| *Tower ~120 m* | High-vol sampler | PM$_{2.5}$ filter samples | IAP | |
| | Anderson sampler | Size-resolved PM samples | IAP | |
| *Tower ~260 m* | High-vol sampler | PM$_{2.5}$ filter samples | IAP | |
| | Anderson sampler | Size- resolved PM samples | IAP | |
| | ACSM | NR PM$_1$ species | IAP | Sun et al. (2012) |
| | CAPS-PM-$_{Ext}$ (630nm) | Extinction | IAP | Wang et al. (2015b) |
| | SMPS 3938 | Particle Number size distribution | IAP | Du et al. (2017) |
| | Gas analyser | CO, O$_3$ and SO$_2$ | IAP | Zhou et al. (2018) |
| | Aethalometer AE33 | Black carbon | IAP | Xie et al. (2018) |
| | Single particle sampler | Individual particles | CUMTB | Wang et al. (2018) |
| **Tower and tower basket measurements** | SNAQ boxes (x 6 at different heights) | CO, NO, NO$_2$, SO$_2$, PM$_1$, PM$_{10}$, PM$_{2.5}$ | Cambridge | Popoola et al. (2018) |
| | LOPAP | HONO (3 min avg) | Birmingham | Crilley et al. (2016) |
| | Spectral radiometer | Photolysis rates | Leeds | Bohn et al. (2016) |
| | SNAQ | CO, NO, NO$_2$, SO$_2$, PM$_1$, PM$_{10}$, PM$_{2.5}$ | Cambridge | Popoola et al. (2018) |
| | WIBS | Fluorescent biological aerosol particles (FBAP) | IAP | Yue et al. (2016) |
| | AE33 | BC | IAP | Xie et al. (2018) |
| | Los Gatos NH$_3$ Analyzer | NH$_3$ | IAP | Meng et al. (2018) |
| | PAX | Light scattering / absorption | IAP | Xie et al. (2018) |
| *IAP groun* | High-Vol sampler | *PM$_{2.5}$ filter samples* | Peking | |





| | 4-channel sampler | *PM$_{2.5}$ filter samples* | Peking | |
| | High Vol sampler | *High time resolution PM$_{2.5}$* filter samples | York | |
| | FDMS+Thermo Scientific 1405-DF | Online PM$_{2.5}$ mass conc. | IAP | |
| | Partisol sampler | PM$_{2.5}$ + PM$_{2.5-10}$ | Birmingham | Taiwo et al. (2014) |
| | Streaker sampler | Hourly elements in PM$_{2.5}$ and PM$_{2.5-10}$ | Birmingham | Taiwo et al. (2014) |
| | Digitel High Vol | PM$_{2.5}$ daily | IAP | |
| | Digitel High Vol | PM$_1$ - 3 hourly | IAP | |
| | Andersen sampler | Size resolved PM | IAP | |
| *IAP roof/lab* | WIBS | Fluorescent biological particles | IAP | Yue et al. (2016) |
| | CAPS-NO$_2$ | NO$_2$ | IAP | Ge et al. (2013) |
| | Aethalometer AE33 | Black carbon | IAP | Xie et al. (2018) |
| | CAPS-PM$_{SSA}$ (630nm) | Extinction, Scattering | IAP | Han et al. (2017) |
| | HR-ToF-AMS | NR-PM species | IAP | Sun et al. (2016) |
| | SP-AMS | Refractory BC and coated aerosol composition | | Wang et al. (2017a) |
| | Iodide FIGAERO-ToF-CIMS | Particle and gas phase molar molecule | IAP | Zhou et al. (2018) |
| | Single particle sampler | Individual particles | CUMTB | Wang et al. (2018) |


Institution names: AIOFM = Anhui Institute of Fine Optics and Mechanics; BNU = Beijing Normal
University; CEH = Centre for Ecology and Hydrology; CUMTB = China University of Mining and
Technology (Beijing); GIG = Guangzhou Institute of Geochemistry, Chinese Academy of Sciences;
NUIST = Nanjing University of Information Science &Technology; IC-CAS = Institute of
Chemistry, Chinese Academy of Sciences
[+] Deployment of instruments both campaigns unless: 10/11/2016 to 25/6/2017
[*] Winter campaign only
[x] OH wave and OH chem refer to the method used to obtain the background signal for the FAGE
instruments which are equipped with a scavenger inlet



**Table 2:** Haze periods during the summer and winter campaign periods.

| Event | Time | PM$_{2.5}$ (µg m$^{-3}$) | Visibility (km) |
|---|---|---|---|
| Winter Haze Event 1 | 11/08 21:00- 11/10 16:00 | 158 (79 - 229) | 4.1 (2.3-8) |
| Winter Haze Event 2 | 11/15 21:00- 11/19 08:00 | 143 (56 - 244) | 4.2(0.6-8) |
| Winter Haze Event 3 | 11/24 12:00- 11/27 02:00 | 210 (68-363) | 4.2(1.5-8) |
| Winter Haze Event 4 | 12/02 16:00- 12/05 02:00 | 239 (58 -530) | 3.9(0.9-8) |
| Winter Haze Event 5 | 12/06 09:00- 12/08 10:00 | 144 (64 -229) | 4.6(2.2-8) |
| Summer Haze Event 1 | 27/05 12:00 -28/05 13:00 | 107(62- 163) | 6.8(4.5-9) |
| Summer Haze Event 2 | 17/06 09:00-18/06 17:00 | 90.5(60-153.3) | 9.3(7-13) |

Note: data in parentheses show the range





**Table 3:** Overview of measurements at the Pinggu site.

| Instruments | Measurements | Insitutue | Reference |
| --- | --- | --- | --- |
| Thermo gas analysers | $NO_x$/$SO_2$/CO/$O_3$ | Peking | Liang et al., 2017 |
| BAM 1020 | $PM_{2.5}$ mass concentration | Peking | Liang et al., 2017 |
| High vol sampler | $PM_{2.5}$ samples | IAP | Zhao et al., 2018 |
| Medium vol sampler | $PM_{2.5}$ samples | IAP | Zhao et al., 2018 |
| Low vol Andersen sampler | Size resolved PM samples | IAP | Zhao et al., 2018 |
| Partisol sampler | $PM_{2.5}$ samples | Birmingham | Taiwo et al. (2014) |
| Streaker sampler | Hourly elements in $PM_{2.5}$ and $PM_{2.5-10}$ | Birmingham | Taiwo et al. (2014) |
| High vol sampler | Filters of $PM_{2.5}$; high time resolution | Birmingham | |
| Four Channel sampler | $PM_{2.5}$ samples | Peking | Liang et al., 2017 |
| Thermo MAAP | Online Black Carbon | Peking | Lin et al., 2011 |
| Sunset OC/EC analyzer | Online OC/EC | Peking | Han et al., 2014 |
| Xact | Hourly metals | Peking | Yu et al. (2018) |
| TOF-ACSM | NR-chemical composition (summer) | Peking | Sun et al., 2012 |
| Thermo Metone | Meteorological parameters | Peking | Liang et al., 2017 |
| SNAQ | Meteorological parameters | Cambridge | Popoola et al. (2018) |
| SP-AMS | Individual particle composition | CQIGIT | Chen et al. (2017) |
| SMPS | Size distribution | Tsinghua | Wang et al., 2009 |
| ACSM | NR-chemical composition (winter) | Tsinghua | Li et al. (2016) |

CQIGIT = Chongqing Institute of Green and Intelligence Technology, Chinese Academy of
Sciences





**Table 4:** Average air quality variables at IAP, Pinggu and 12 national monitoring sites (12N)
during the field campaigns (10 Novemberember – 11 December 2016; and 21 May – 22 June 2017).
The 12 national sites five-year mean concentrations for same times of the years (12N -5Y) and for
the same time of the year (campaign period) (12N-campaign).  Data are mean ± s.d. (range).

| Pollutant[1] | Winter (10 Nov-11 Dec 2016) | | | | Summer (21 May-22 June 2017) | | | |
|---|---|---|---|---|---|---|---|---|
| | IAP | PG | 12N-5Y | 12N - campaign | IAP | PG | 12N-5Y | 12N- campaign |
| $PM_{2.5}$ [2] | 91.2 ± 63.7 (10.3-239.9) | 99.7 ± 77.8 (13.3-294.3) | 84.01 ± 89.1 (3.2-593.3) | 95.3 ± 79.6 (4.7-408.8) | 31.4 ± 14.7 (12.2-78.8) | 27.8 ± 13.3 (10.6-70.3) | 58.7 ± 40.0 (4.2-250.3) | 41.7 ± 22.3 (8.9-134.1) |
| $PM_{10}$ [2] | 130.6 ± 87.0 (20.0-329.2) | 121.9 ± 80.4 (10.4-312.1) | 112.8 ± 102.2 (5-662.0) | 134.5 ± 100.4 (6.0-550.1) | 74.9 ± 29.3 (22.5-164.6) | 62.9 ± 29.3 (15.1-141.9) | 94.6 ± 52.7 (5.0-463.2) | 81.9 ± 37.1 (6.0-277.8) |
| $NO_2$ | 69.7 ± 33.3 (10.2-167.3) | 46.4 ± 25.5 (2.3-132.4) | 57.7 ± 33.9 (3.9-166.4) | 66.4 ± 31.3 (7.3-156.6) | 41.3 ± 23.5 (9.2-142.9) | 29.3 ± 10.3 (9.3-84.0) | 40.6 ± 17.9 (8.1-132.4) | 37.6 ± 16.2 (12.5-92.8) |
| $SO_2$ | 14.9 ± 11.1 (0.1-50.8) | 15.4 ± 6.7 (6.2-44.4) | 16.6 ± 16.2 (1.4-112.0) | 14.2 ± 9.4 (2.1-51.4) | 6.3 ± 6.8 (0.1-38.2) | 8.9 ± 4.7 (4.2-41.2) | 10.1 ± 10.6 (1.8-82.3) | 7.4 ± 6.6 (1.8-64.5) |
| $CO$ [2] | 1.53 ± 1.02 (0.7-5.0) | 1.47 ± 1.17 (0.1-6.9) | 1.65 ± 1.38 (0.1-9.6) | 1.86 ± 1.17 (0.3-5.7) | 0.61 ± 0.32 (0.1-2.5) | 0.52 ± 0.29 (0.1-2.3) | 0.93 ± 0.74 (0.2-8.7) | 0.74 ± 0.33 (0.2-2.5) |
| $O_3$ | 16.4 ± 17.0 (0.3-63.3) | 22.3 ± 22.2 (2.9-78.0) | 21.8 ± 20.5 (1.0-72.9) | 17.5 ± 19.2 (2.1-67.4) | 106.9 ± 71.6 (2.0-349.3) | 91.8 ± 62.7 (0.2-291.4) | 100.4 ± 67.8 (2.2-343.5) | 110.8 ± 66.5 (3.6-335.9) |

[1], Units: µg m$^{-3}$ except CO units: mg m$^{-3}$
[2], $PM_{2.5}$ and $PM_{10}$ from IAP and Pinggu measured by a gravimetric method; all other data are online
measurements hourly mean.





**Table 5:** Mean and standard deviation (sd) of climatological conditions in Beijing for each circulation type (CT) for 1988-20 17 from Era Interim data with frequency of the CT during the **W** (winter) and **S** (summer) campaigns (% of 6 h periods (p)) compared to A- long- term 1988-2017.

| CT | Description | WS m s$^{-1}$ | WS$_{sd}$ m s$^{-1}$ | WD ° | WD$_{sd}$ ° | T2m °C | T2m$_{sd}$ °C | TD2m °C | TD2m$_{sd}$ °C | MSLP hPa | MSLP$_{sd}$ hPa | RH % | RH$_{sd}$ % | Season | Frequency (%) W | S | A |
|---|---|---|---|---|---|---|---|---|---|---|---|---|---|---|---|---|---|
| 1 | H - west of the domain | 3.38 | 1.63 | 298.3 | 62.6 | 0.1 | 7.1 | -12.6 | 7.9 | 1026.50 | 4.14 | 41 | 18 | Winter monsoon | 16 | 7 | 9.3 |
| 2 | H - west of the domain | 2.91 | 1.49 | 265.9 | 107.0 | -2.8 | 6.2 | -13.8 | 7.5 | 1034.34 | 4.47 | 45 | 18 | Winter monsoon | 1 | 0 | 7.2 |
| 3 | relatively L in NE further reduction | 3.21 | 1.65 | 281.2 | 71.3 | 6.8 | 8.9 | -6.4 | 9.3 | 1017.77 | 4.35 | 43 | 20 | Sep- May | 12.5 | 0 | 8.3 |
| 4 | L (cf. CT3, 5) in NE winds start to turn over Yellow Sea | 3.05 | 1.73 | 240.1 | 104.1 | 19.2 | 7.5 | 7.0 | 10.4 | 1007.20 | 3.63 | 50 | 24 | Mar-Aug Spring - summer | 11.8 | 4 | 7.8 |
| 5 | relatively L in NE | 2.57 | 1.37 | 189.1 | 125.0 | 8.2 | 8.9 | -0.9 | 10.4 | 1020.82 | 4.62 | 57 | 23 | Sep-May | 7.6 | 34 | 8.3 |
| 6 | further reduction L (cf. CT3, 5) in NE | 2.58 | 1.32 | 197.4 | 87.6 | 24.6 | 5.9 | 14.7 | 8.0 | 1000.99 | 2.96 | 59 | 23 | Summer monsoon | 8.3 | 12 | 8.9 |
| 7 | when winds are oriented westward from the Bohai Sea | 2.29 | 1.12 | 167.5 | 100.2 | 18.9 | 7.8 | 10.7 | 9.5 | 1012.59 | 3.61 | 63 | 21 | | 1 p | 11 | 10.2 |
| 8 | like CT 6 | 2.35 | 1.11 | 165.4 | 75.4 | 24.0 | 5.3 | 15.9 | 6.8 | 1006.47 | 2.69 | 65 | 21 | Summer monsoon | | 32 | 12.9 |
| 9 | Air mass stagnant over Beijing | 2.03 | 0.94 | 208.7 | 107.4 | 2.1 | 7.9 | -6.2 | 8.4 | 1028.66 | 4.18 | 58 | 20 | | | 0 | 9.6 |
| 10 | Air mass stagnant over Beijing | 2.67 | 1.17 | 211.1 | 68.7 | 14.2 | 9.4 | 3.1 | 10.0 | 1013.98 | 3.84 | 52 | 22 | | 25 | 0 | 7.2 |
| 11 | Air mass stagnant over Beijing | 2.23 | 0.98 | 209.1 | 86.5 | 8.1 | 9.4 | -0.4 | 9.6 | 1021.83 | 4.06 | 59 | 20 | | 16 | 0 | 10.3 |

Note: WS- wind speed, WD wind direction, T2m – 2 m air temperature, TD2m – 2 m dewpoint temperature, MSLP – mean sea level pressure, RH – relative humidity; L – low pressure; H – High pressure









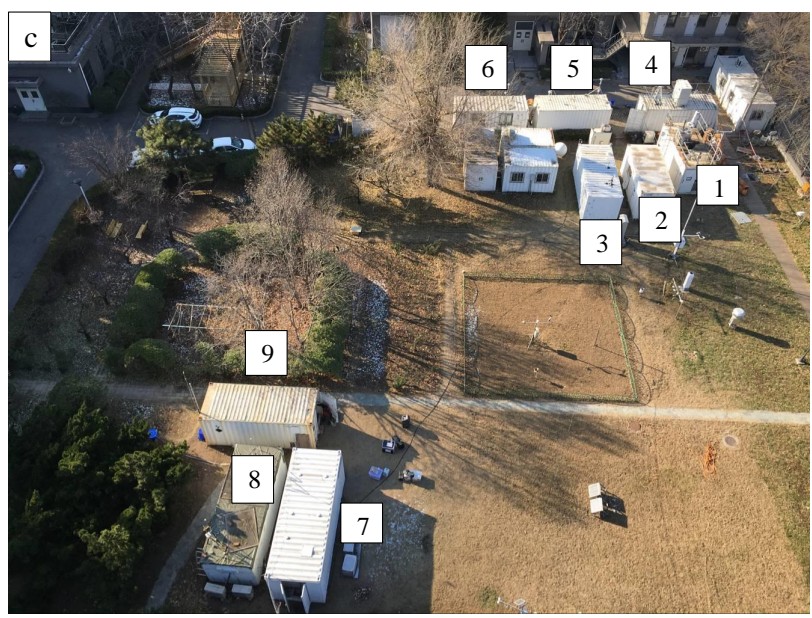

1265

**Figure 1:** Study area topography (source: googlemap) of Beijing / Tianjing / Hebei region (a) with the rectangle showing enlarged study area; locations of measurement sites (Institute of Atmospheric Physics (IAP)– urban Beijing, Pinggu – rural Beijing; and Gucheng – upwind site in Hebei province), SNAQ box sites (red symbols) and the 12 national air quality monitoring stations (G1 to G12, blue symbols) (b); locations of the 9 containers at IAP (c) – instrumentation at each container is shown in Table 1. The shaded area shows the Beijing buildup area. (Source: a and b - Goggle Map topographic background imagery; c – taken by ~~Siyao Yue~~ Jian Zhao from IAP).

G1: Wangshouxigong; G2: Dingling; G3: Dongsi; G4: Tiantan; G5: Nongzhanguan; G6: Guanyuan; G7: Haidianquwanliu; G8: Shunyxicheng; G9: Huairouzhen; G10:Changpingzhen; G11: Aotizhongxin (Olympic Park); G12: Gucheng. Categories: Urban: G1, G3, G4, G5, G6, G7, G8, G11, G12; Suburban: G9, G10; Rural: G2.





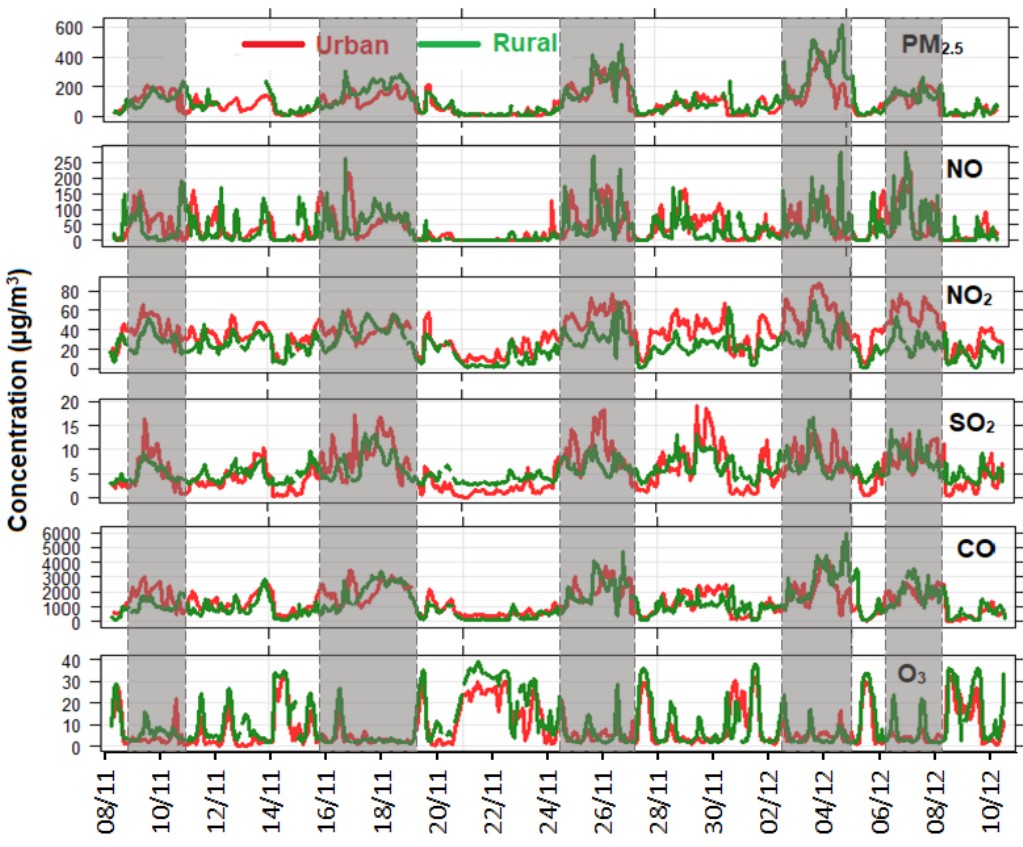


**Figure 2:** Time-series of air quality variables at the urban and rural sites during the winter
campaign; Five haze events are indicated (shading).







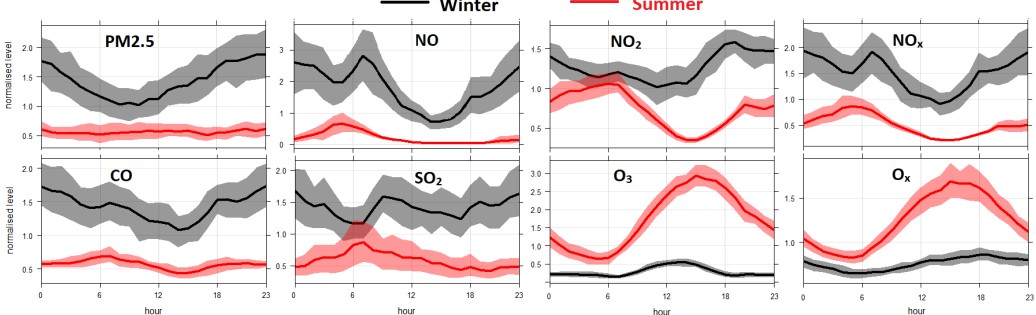


**Figure 3:** Diurnal patterns of gaseous pollutants normalized by average concentrations at IAP
during winter and summer campaigns. Line shows the mean concentrations and shaded area as 95%
confidence interval in the difference in mean concentrations.





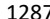




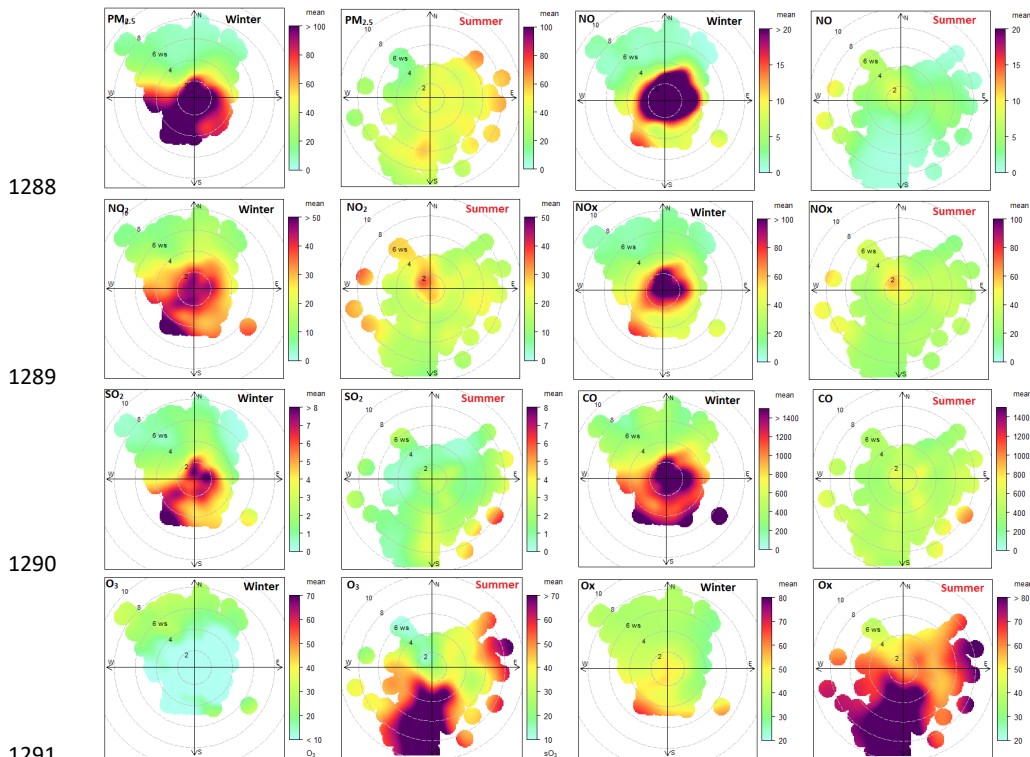


**Figure 4:** Air pollutants concentration (colour) with wind direction (angle) and wind speed (m s⁻¹)
at IAP during the winter and summer campaigns. Data are hourly in time resolution and were from
10 November to 11 December 2016 (winter) and 21 May to 22 June 2017 (summer). The colour
scale is for "weighted.mean" where the mean wind speed/direction bin is multiplied by the bin
frequency and divided by total frequency.






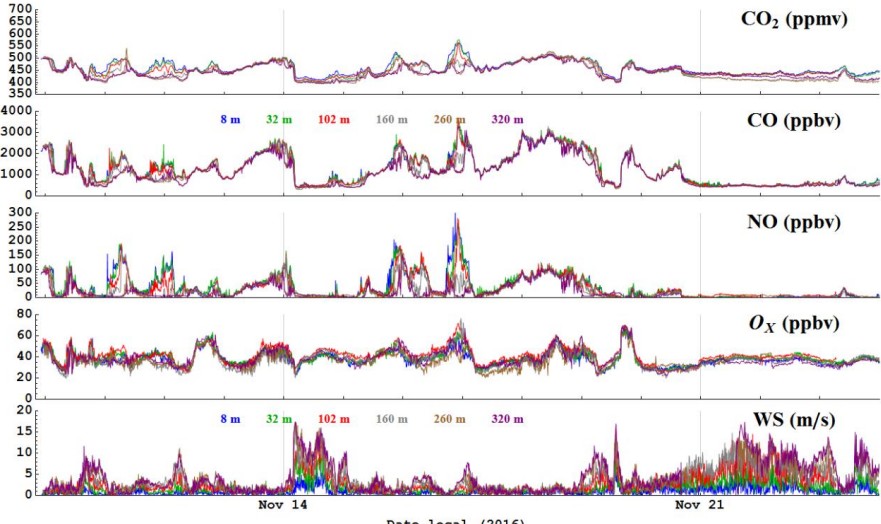


**Figure 5:** Time series of $CO_2$, CO, NO, $O_x$ ($NO_2+O_3$) and wind speed at six heights (colour)
measured with SNAQ boxes on the IAP tower during the winter intensive field campaign.






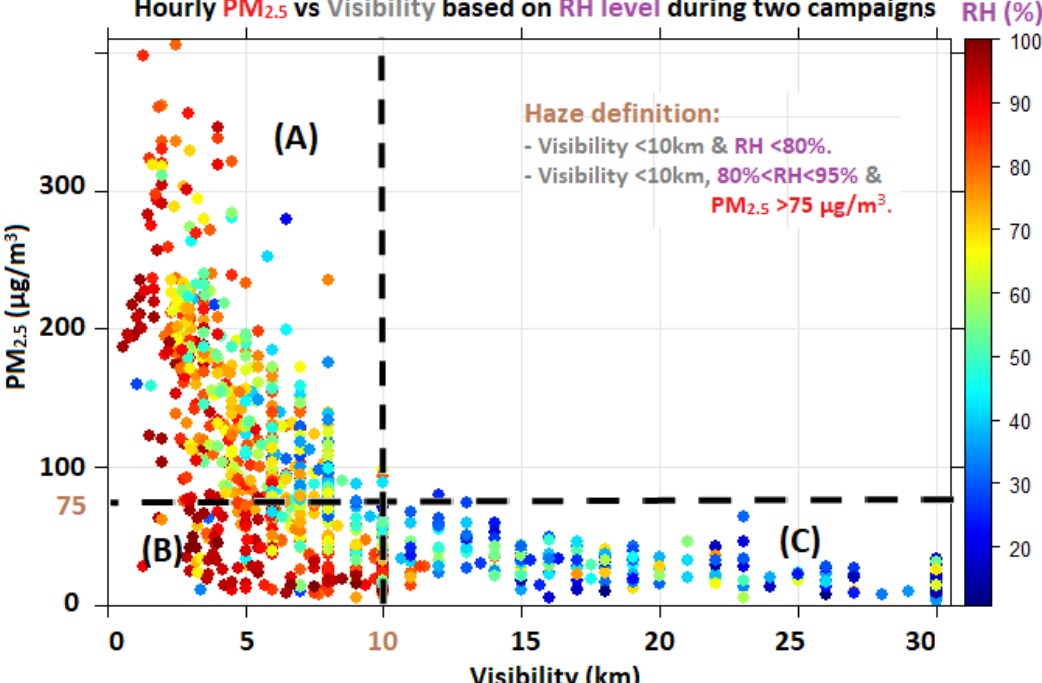


**Figure 6:** Hourly PM$_{2.5}$ mass concentrations versus visibility (at the Beijing Capital Airport) during
the winter campaign. Data source: visibility downloaded using R-"worldmet" package: date of last
access: 27/02/2018).





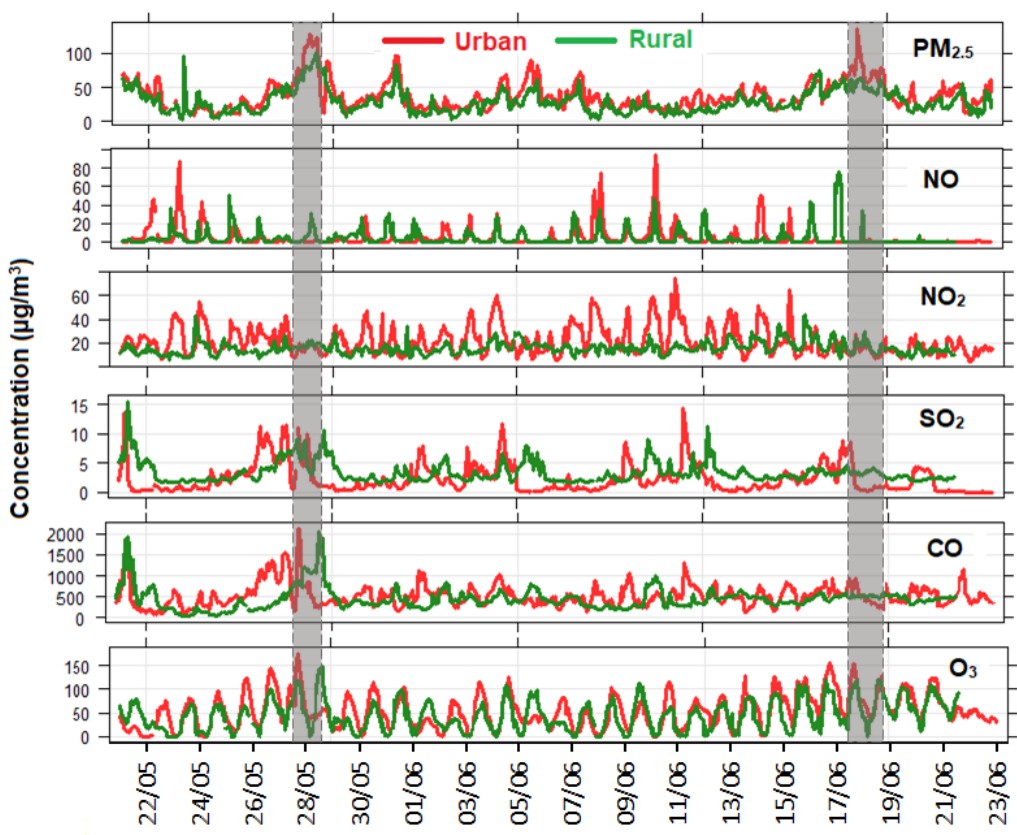

**Figure 7:** Time-series of air quality variables at the urban and rural sites during the summer
campaign. Two minor haze events are indicated (shading).





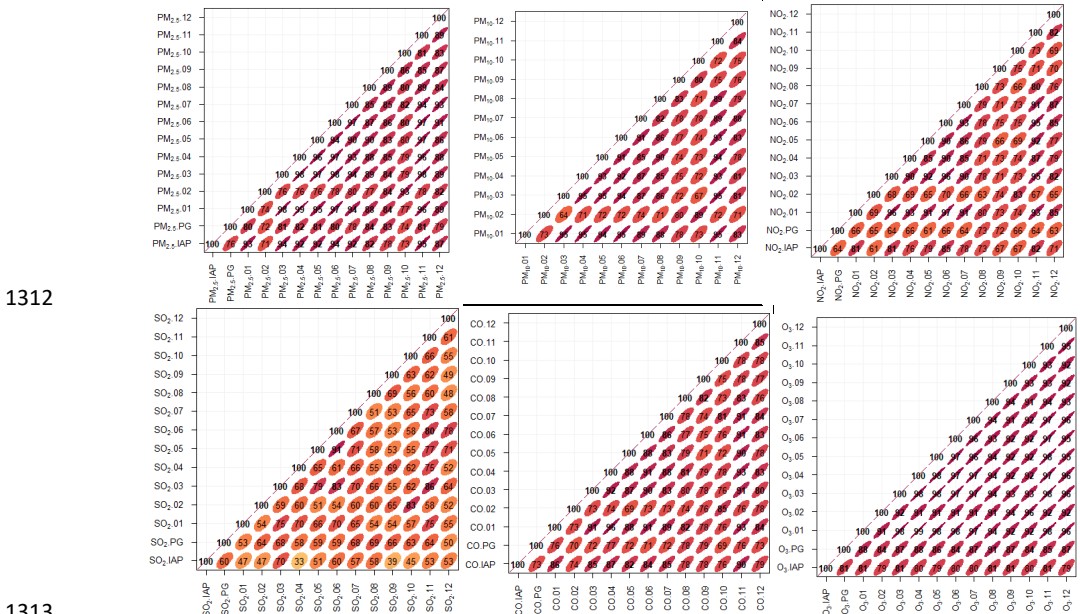

**Figure 8:** Correlations between the air quality at IAP, PQ and 12 monitoring station around
Beijing. Stations G1-G12 (Figure 2) are labelled 01-12, PG = Pinggu.





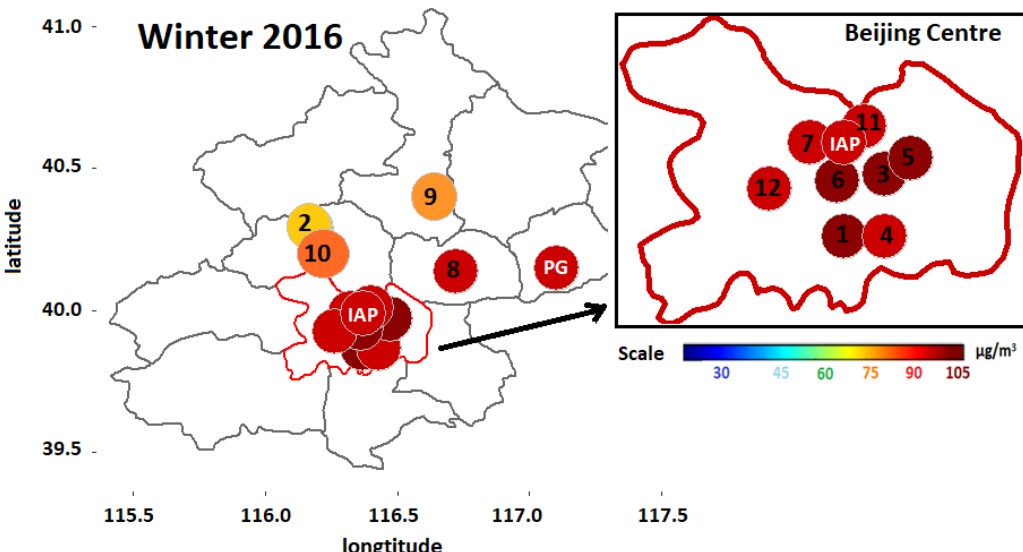



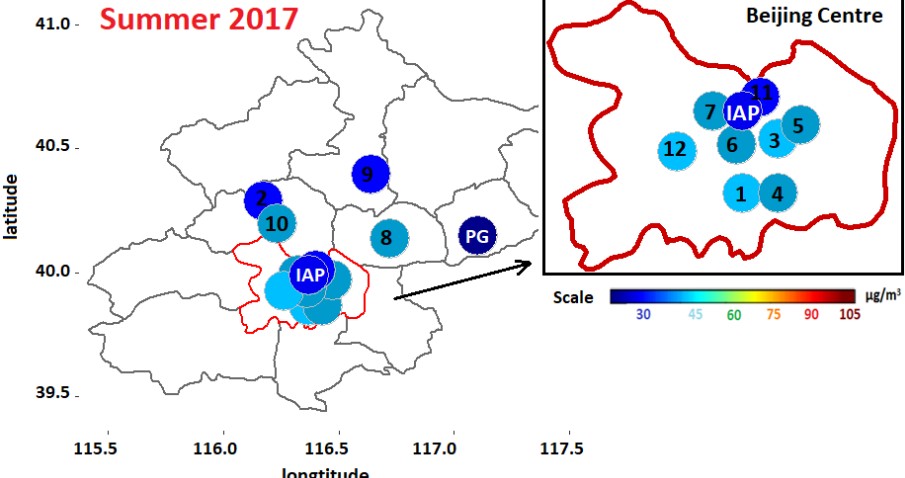


**Figure 9:** Spatial distribution of hourly mean concentration of PM$_{2.5}$ in Beijing during two
sampling campaigns.







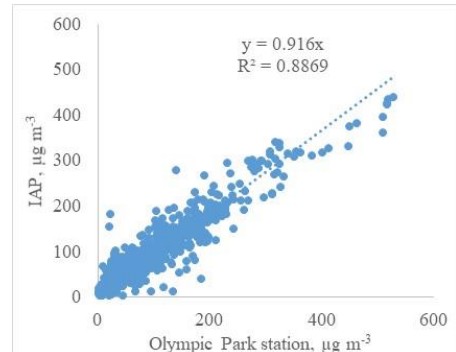


**Figure 10:** Hourly $PM_{2.5}$ at IAP (roof of a two storied building) and the neighbouring Olympic park
national air quality monitoring station during the winter and summer intensive field campaigns.






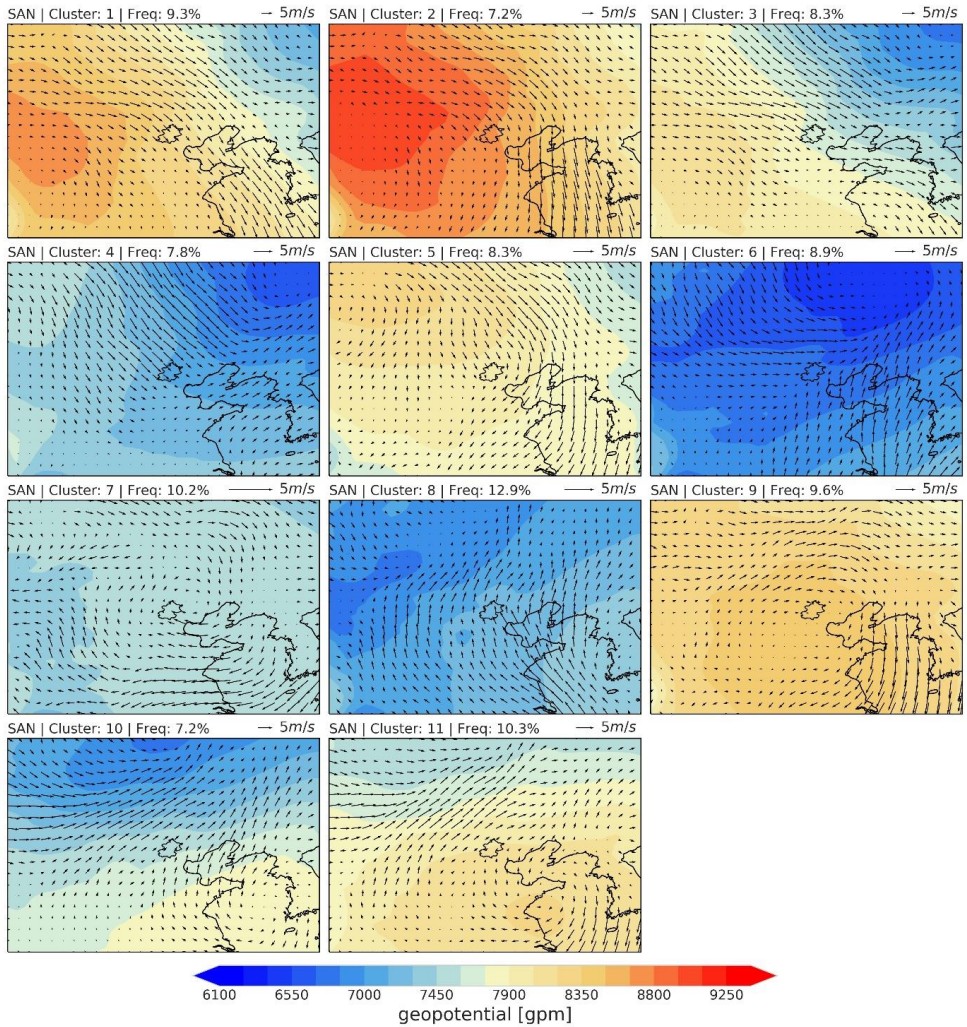

**Figure 11:** ERA-Interim (1988-2017) average 925 hPa geopotential with 10 m horizontal wind vector for 11 circulation types classified for Beijing (municipal boundary thin solid line) surroundings (103-129° E, 31 - 49° N) determined with the SANDRA method (COST733 class software). Frequency of occurrence is given in cluster caption. For discussion of conditions associated with each CT see section 6.1.






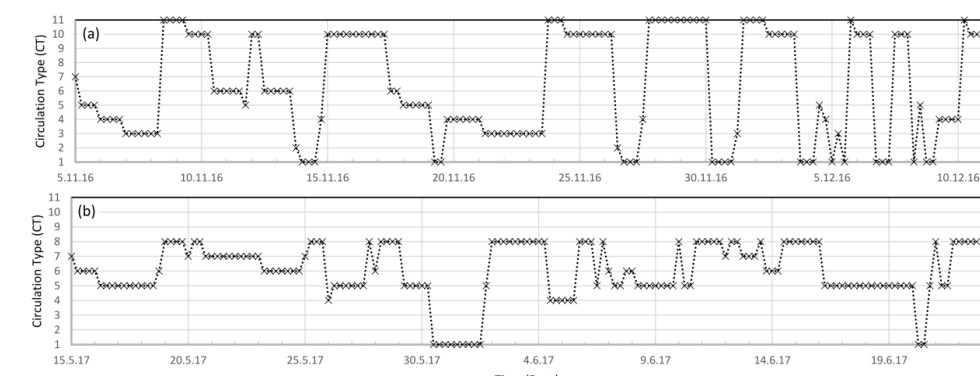


**Figure 12:** Time series of circulation types (CTs) during the two field campaigns: (a) winter and (b)
summer. The 11 CTs are shown in Figure 11. See text for more description.




**Figure 13**: Analysis by circulation type (CT; Sect. 0) of: (a) daily maximum mixed layer height
(MLH) determined from ALC observations at IAP between November 2016 – June 2017 (analysis
method, Kotthaus and Grimmond, 2018b); concentration of (b) PM$_{2.5}$ and (c) O$_3$ at at the Olympic
Park (i.e. Aotizhongxin) in 2013-2017 from the national air quality network; occurrence of CTs in
(d) 1988-2017 and (e) Oct 2016 – Sept 2017; (f) anomaly of CT frequency during Oct 2016 – Sept
2017 compared to 30 y climatology; and (g) anomaly of PM$_{2.5}$ and O$_3$ during Oct 2016 – Sept 2017
compared to 5 y (2013-2017) average (same data as in b, c).





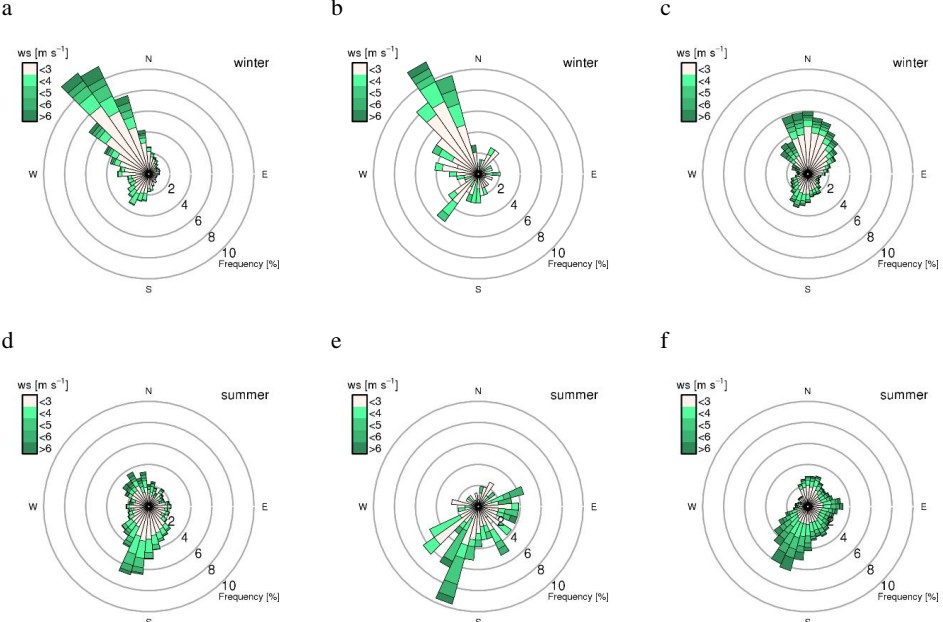

1351

**Figure 14:** Beijing wind roses: (a, b, d, e) ERA-Interim 10 m horizontal wind (40° N, 116.5° E) and (c, f) sonic anemometer (Table 1) at IAP 320 m agl for (a) 5 November – 10 December in 1988-2017, (d) 15 May – 22 June in 1988-2017, (b, c) 5 November – 10 December 2016, and (e, f) 15 May – 22 June 2017.









**Figure 15:** Hourly meteorological variables measured at 120 m during the (a) winter and (b)
summer campaigns. The shaded areas highlighted the haze periods (Table 3, Figures 2 and 7).







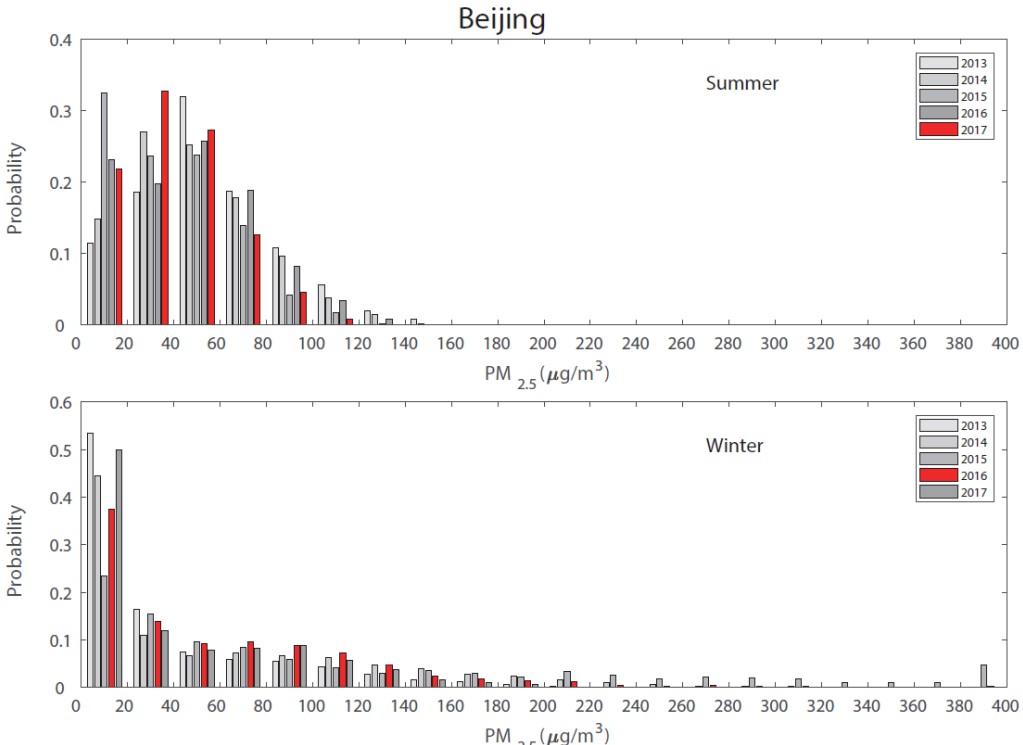


**Figure 16**: Frequency distribution of PM$_{2.5}$ in Beijing over the summer (top) and winter (bottom)
campaign periods from the NAQPMS model compared with those from the same periods over the
past five years under the same emission conditions
