# Peer review of "Introduction to Special Issue - In-depth study of air pollution sources and processes within Beijing and its surrounding region (APHH-Beijing)"

_Atmospheric Chemistry and Physics, 2018_

## Referee Comment (RC1) · Anonymous Referee #1 · 14 Nov 2018

As an introduction to a special issue, the manuscript should be improved better. Honestly speaking, this reviewer has difficulties to follow many parts. Moreover, several giant projects on air pollution and health impacts funded in the volume of billions RMB yuan are processing currently or have been completed in Beijing and neighboring provinces in the last decade. These works should be summarized to make the literature review more complete. The authors are strongly encouraged to present a summary to highlight the importance of APHH-Beijing in comparison with others. Section 2 is too ambitious to be practical for two short-term campaigns. Please consider to revise. This reviewer also has several minor comments for authors considering.

[Figure]

1) Lines 76-80 "The winter campaign was characterized by high PM2.5 pollution events whereas the summer experienced high ozone pollution events. Air quality was poor during the winter campaign, but less severe than in the same period in 2015 when there were a number of major pollution episodes. PM2.5 levels were relatively low during the summer period, matching the cleanest periods over the previous five years." The statement looks like the report issued by local EPD rather than a scientific study. The reviewer gains almost nothing from it. It should be more specific.

2) Lines 80-82, "Synoptic scale meteorological analysis suggests that the greater stagnation and weak southerly circulation in November/December 2016 may have contributed to the poor air quality." Contributed to a few or all severe PM2.5 pollution events?

3) Line 100, "particularly severe in developing megacities, such as Beijing, where rapid urbanisation has led to a fast increase in pollution emissions (Guan et al., 2014), on top of regional pollution from industrial and other anthropogenic activities." Can Beijing be called as developing megacities? The reviewer also cannot understand the statement, please consider to revise.

4) Lines 117-119 "This makes Beijing a particularly interesting place to study as it provides a new environment to test our understanding of urban pollution processes." The reviewer feels very surprised that all Chinese co-authors agree with the statement.

5) The objectives in Section 2 are ambitious. The reviewer has doubt how they can be achieved through two short-term campaigns at two sites.

6) Section. 3.1.1 does not sound scientific to this reviewer. It makes more sense to use the data from the air quality monitoring network and the two additional sites together to evaluate the accuracy of emissions of air pollutants?

7) Lines 243-246" Previous studies of pollution in Beijing have shown that it is often perturbation of the physicochemical and dynamic atmospheric conditions that modulate the most severe air quality events, rather than changes in emissions, for example during the development of stable inversions or periods of strong photochemistry." The references are missing. Please consider to revise. The statement is hard to follow.

8) Lines 280-283, " AIRLESS aimed to advance air quality and health research in China by bringing together two fields of research that have made rapid advancements in recent years: measurements of a wide range of pulmonary and cardiovascular biomarkers in a panel study and personal monitoring of multiple air pollutants with high spatio-temporal resolution by sensor technology" In China or In Beijing and Neighboring Provinces? Why are the two sites' measurements helpful for the targets?

9) This reviewer listed only a few. The authors are encouraged to check carefully for the remaining parts.

---

## Referee Comment (RC2) · Anonymous Referee #2 · 14 Nov 2018

This manuscript provides an overview of the APHH-Beijing project ïĆ¿ the objectives, the research themes, the measurement and modeling activities, the fundamental air quality data, and so on. The authors spend a large body of the main text describing in detail the background, justification, and the to do lists of this project, making this manuscript more like a white paper or a proposal draft. I understand that the purpose of this manuscript is to provide a starting point and an introduction for readers who are interested in many future publications that are likely to result from this international collaborative program. However, to be qualified as an ACP research article, the authors need to provide more sciences in the manuscript, as suggested below.

[Figure]

1. There have been many field campaigns, e.g., CAREBeijing, organized in the past 10 years targeted on investigating the air pollution and its health impact in Beijing. Compared with all those previous studies, what is unique about the current project? What are the scientific challenges this project aims to solve?

2. There are four research themes presented: sources and emissions, atmospheric processes, health effects, and solutions. The last two only appeared in the very first part of the manuscript, no scientific output can be found later on. To make this manuscript completer and more consistent, primary results related to the health impacts need to be given.

3. One focus of this manuscript is the overview of two joint field campaigns. Indeed, there are lots of discussions regarding the site information and type of instruments, but these discussions are not necessarily useful, as any future publications related to these two campaigns would have to give similar descriptions in their methods section anyway. Instead, this manuscript could be a nice platform for a detailed instrument calibration and comparison, data analysis and uncertainty quantification, and so on.

4. The last two sections describe the air quality, e.g., the average concentrations and diurnal patterns of common air pollutants like NOx, O3, PM2.5, and etc., during the two field campaigns. As the authors highlighted earlier that regional modeling is an essential part of the campaigns, a modeling vs. observation comparison in terms of temporal profiles of these common pollutants need to be provided.

---

## Author Response (AR1)

**Response to reviewer comments:**

**General response:**

We thank both reviewers for providing constructive comments.

We have carefully considered every single comment and revised the manuscript accordingly. We also provided a point by point response to all the comments made below.

One point we would like to make is that this is an Introduction to special issue paper, not an overview or research paper. ACP editorial policy states that "*Special issues may include an introduction article or an overview article or both. Introduction articles* **outline the motivation and background**, *and overview articles synthesize and summarize the findings of the special issue papers. The manuscript title must clearly reflect the relation to the special issue and should start with "Introduction:" or "Overview*".

To make this clearer, we have added a paragraph at the end of the introduction.

*"This introduction paper describes the motivation and background of APHH-Beijing programme, and presents some of the background air quality and meteorology observations that lay the basis of data interpretation for the whole programme, particularly during the two intensive field campaigns. These campaigns form one of the core research activities within APHH-Beijing integrating the different themes / projects. We did not intend to present the key scientific results of APHH-Beijing here as much of the research activities are still ongoing and unpublished. Such information is more suitable to go to an overview paper."*

We also would like to emphasize that scientific work on the impact of synoptic scale meteorology on air quality and the air quality climatology add significant knowledge to our understanding of air pollution events in Beijing. Therefore, this introduction paper not only provides the motivation and background of the APHH-Beijing programme but also new science.

Many of the ACP special issues have introduction papers, such as:

Kulmala, M. et al., 2009. Introduction: European Integrated Project on Aerosol Cloud Climate and Air Quality interactions (EUCAARI) – integrating aerosol research from nano to global scales. Atmos. Chem. Phys., 9, 2825–2841.
Cairo, F., et al., 2010. An introduction to the SCOUT-AMMA stratospheric aircraft, balloons and sondes campaign inWest Africa, August 2006: rationale and roadmap. Atmos. Chem. Phys., 10, 2237–2256
Kruger, K and Quack, B., 2013. Introduction to special issue: the TransBrom Sonne expedition in the tropical West Pacific. Atmos. Chem. Phys., 13, 9439–9446
Kulmala, M. et al., 2015. Introduction: The Pan-Eurasian Experiment (PEEX) – multidisciplinary, multiscale and multicomponent research and capacity-building initiative. Atmos. Chem. Phys., 15, 13085–13096
Martin, S.T. et al., 2016. Introduction: Observations and Modeling of the Green Ocean Amazon (GoAmazon2014/5). Atmos. Chem. Phys., 16, 4785–4797.

**Reviewer 1**

*Comment 1*: several giant projects on air pollution and health impacts funded in the volume of billions RMB yuan are processing currently or have been completed in Beijing and neighboring provinces in the last decade. These works should be summarized to make the literature review more complete. The authors are strongly encouraged to present a summary to highlight the importance of APHH-Beijing in comparison with others.

*Response*: We agree that a summary of past work in Beijing will be valuable to put the APHH-Beijing work into context and we have added a summary about the CARE-Beijing and other large programmes (see below). APHH-Beijing programme was designed in 2015 and started in 2016. The rationale of this programme in the introduction paper was based on work up to 2016. Thus, we feel that it is not totally appropriate to include ongoing and unpublished work in this introduction paper.

*Changes in the texts*: We have added texts in the Introduction

*"Many research programmes were initiated in Beijing to study the air pollution processes since late 1990s. Earlier research programmes (e.g., early 2000) focused on primary emissions of SO2, NO2, CO, PM10, volatile organic compounds and then secondary pollutants such as ground-level ozone and secondary fine particles. These researches contributed to the development of air pollution mitigation strategies by the Beijing Municipal government.*

*Beijing Olympic Games (2008) offered additional incentives to improve air quality and this led to the funding of CAREBEIJING (Campaigns of Air Pollution Research in Megacity Beijing and Surrounding Region) and other major programmes. CAREBEIJING was initiated and organized by Professor Tong Zhu of Peking University, with participation of hundreds of scientist and students from China, USA, Germany, Italy, Japan, and South Korea. The field campaigns were conducted in the summer of 2006, 2007, and 2008, with the objectives to learn the environmental conditions of the region, to identify the processes (transport and transformation) that lead to the impact of the surrounding area on air quality in Beijing, to quantify the impact of the surrounding area on air quality in Beijing, and to formulate policy suggestion for the air quality attainment during the 2008 Beijing Olympic Games. Other major research programmes, initiated since early 2000, aimed to provide scientific basis to deliver air pollution mitigation measures for ensuring a good air quality during the Olympics Games. Measures developed as a result of these programmes successfully reduced the air pollution during the Olympics Games, and provided valuable examples for air pollution control policy-making in other cities (Wang et al., 2010). CARE-BEIJING latter on was extended to CAREBEIJING-NCP (Campaigns of Air Pollution Research in Megacity Beijing and North China Plain), where field campaigns were carried out in the summer of 2013 and 2014 to investigate the transport and transformation processes of air pollutants in megacity Beijing and North China Plain. The results of CAREBEIJING and CAREBEIJIN-NCP have been published in three special issues of Atmospheric Chemistry and Physics (https://www.atmos-chem-phys.net/special_issue198.html) and Journal of Geophysical Research-Atmospheres (https://agupubs.onlinelibrary.wiley.com/doi/toc/10.1002/(ISSN)2169-8996.CARBS1). These large research programmes and numerous discovery science projects significantly enhanced our understanding on the emission, sources and processes of air pollutants in Beijing (Chan and Yao, 2008; Zhu et al., 2012). However, our understanding of sources and emissions of key air pollutants such as PM2.5 and ozone and the role of the interactions between physical and chemical processes in the formation of pollution events in the Beijing megacities is still far from being accurate or complete. In addition, none of the abovementioned large programmes are directly linked heath effect studies. "*

In addition, we have added a section at the end of the paper to summerize this introduction paper and highlight the novel aspects of the APHH-Beijing.

*"The APHH-Beijing is an integrated and multidisciplinary research programme by leading UK and China researchers to (1) quantify sources and emissions of urban atmospheric pollutants; (2) elucidate processes affecting urban atmospheric pollution events; (3) estimate the personal exposure and impacts of air pollution on human health, and (4) develop intervention strategies to improve air quality and reduce health impacts in the Beijing megacity. This introduction paper outlines the motivation of the APHH-Beijing programme as well as provides the background air quality and meteorological conditions that form the basis of data interpretation for the whole programme, particularly during the two intensive field campaigns as a core research activity within the programme.*

*APHH-Beijing has measured the fluxes of key air pollutants, including NOx, CO, BC, VOCs and speciated particulate matter, applied a suite of traditional and modern techniques to apportion the sources of particulate matter, determined a wide range of pulmonary and cardiovascular biomarkers linking to direct personal exposure and extensive fixed-station monitoring as well as source apportionment results, and evaluated the effectiveness of Beijing's air pollution control policies using both chemical transport models and novel machine learning techniques. A number of papers have already been published under the APHH-China programme including those in APHH-Beijing special issue (Wang et al., 2019; Pan et al., 2019; Xia et al., 2018; Zhou et al., 2018; Wang et al., 2018; Lyu et al., 2019; Hollaway et al., 2019; Du et al., 2018; Liu et al., 2018a,b; Smith et al., 2018). More papers are being prepared for publication in this special issue and elsewhere, which will cover (but not limited to) emission fluxes of air pollutants, chemical composition and source apportionment of fine particles, satellite observations of trace gases and aerosols, sources and processes leading to haze events and photochemical smogs, physical and optical properties of aerosol particles, formation processes of secondary aerosols, urban meteorology, feedbacks between haze, photochemistry and meteorology, integrated regional and urban scale modelling, personal exposure to air pollutants and human health effects of air pollution."*

**Comment 2**: Section 2 is too ambitious to be practical for two short-term campaigns

**Response**: It appears that there is a misunderstanding here. The whole programme is more than the two campaigns. We have introduced the two campaigns because they are one of the core activities that integrate research across the themes and the information provided in this introduction paper provided a background for a number of in-preparation papers for this special issue.

Section 2 is extracted from the five funded proposals that were awarded on a competitive basis and assessed by international expert reviewers and a panel of UK/China top scientists. Now look back at these set objectives, we have indeed make progresses in all areas.

**Comment 3**: Lines76-80 "The winter campaign was characterized by high PM2.5 pollution events whereas the summer experienced high ozone pollution events. Air quality was poor during the winter campaign, but less severe than in the same period in 2015 when there were a number of major pollution episodes. PM2.5 levels were relatively low during the summer period, matching the cleanest periods over the previous five years." The statement looks like the report issued by local EPD rather than a scientific study. The reviewer gains almost nothing from it. It should be more specific.

**Response**: This is a very general introduction which sets our campaign periods in context. However, we recognise that it is very qualitative and have therefore modified it to include quantitative information which guides the reader more usefully. The revised text reads as follows:

**Changes made**:

*"The winter campaign was characterised by high PM2.5 pollution events with peak hourly concentrations at the urban site ranging up to 498 µg m$_{-3}$, whereas the summer experienced events of high ozone concentrations with the highest hourly average up to 176 ppb.  Air quality was generally poor during the winter campaign with an average PM2.5 concentration of 96 µg m$^{-3}$, but less severe than in the same period in 2015. Synoptic scale meteorological analysis suggests that the greater stagnation and weak southerly circulation in November/December 2016 contributed to the poor air quality during all haze events detected. PM2.5 levels were relatively low during the summer campaign with the highest daily concentration of only 79 µg m$^{-3}$, matching the cleanest periods over the previous five years."*

**Comment 4**: Lines 80-82, "Synoptic scale meteorological analysis suggests that the greater stagnation and weak southerly circulation in November/December 2016 may have contributed to the poor air quality." Contributed to a few or all severe PM2.5 pollution events?

**Response**: We updated Figure 12 to include indication of haze events. This clearly demonstrates that CTs associated with stagnation (CT9, 11) dominate during all haze events of the winter campaign. The text and abstract have been updated accordingly.

The sentence in the abstract is now changed to
*"Synoptic scale meteorological analysis suggests that the greater stagnation and weak southerly circulation in November/December 2016  contributed to the poor air quality during all haze events detected."*

**Comment 5**: Line 100, "particularly severe in developing megacities, such as Beijing, where rapid urbanisation has led to a fast increase in pollution emissions (Guan et al., 2014), on top of regional pollution from industrial and other anthropogenic activities." Can Beijing be called as developing megacities? The reviewer also cannot understand the statement, please consider to revise.

**Response**: We recognize that the definition of a "developing megacity" can sometimes be controversial. In this context, we argue that Beijing is a developing megacity because it is still transforming rapidly and its GDP growth is significantly faster than developed megacities. We have revised the sentence:

**Changes made**:  The quoted sentence has now been changed to:
*"Air pollution is particularly severe in developing megacities, such as Beijing, where pollutants from traditional sources, such as solid fuel combustion are mixed with those from road traffic (Guan et al., 2014), on top of regional pollution from industrial and other anthropogenic activities."*

**Comment 6**: Lines 117-119 "This makes Beijing a particularly interesting place to study as it provides a new environment to test our understanding of urban pollution processes." The reviewer feels very surprised that all Chinese co-authors agree with the statement.

**Response**: We have revised this sentence as following:
*"This makes Beijing a particularly interesting place to study as it provides an atmospheric environment very different to developed megacities such as London and Paris to investigate urban pollution processes."*

**Comment 7**: The objectives in Section 2 are ambitious. The reviewer has doubt how they can be achieved through two short-term campaigns at two sites.

*Response:* see above response to comment 2.

*Comment 8*: Section. 3.1.1 does not sound scientific to this reviewer. It makes more sense to use the data from the air quality monitoring network and the two additional sites together to evaluate the accuracy of emissions of air pollutants?

*Response*: The air quality monitoring network is a valuable source of data but measures only a small suite of classical pollutants whereas our monitoring campaigns measured a much larger range of species, which are helpful in constraining the numerical models. We also made air pollutant flux measurements at the IAP site and that can only be done with a tower. We have also been analysing data from the monitoring network in Beijing and agree that this is a valuable resource in model validation studies. Consequently, we have modified the final paragraph of Section 3.1.1 to read as follows:

*"Measured ground level concentrations both from our campaign sites and the Beijing monitoring network, together with source apportionment results, are compared with the predictions of a chemistry-transport model and used to provide a clear distinction between advected regional pollution and the impact of local sources...".*

*Comment 9*: Lines 243-246" Previous studies of pollution in Beijing have shown that it is often perturbation of the physicochemical and dynamic atmospheric conditions that modulate the most severe air quality events, rather than changes in emissions, for example during the development of stable inversions or periods of strong photochemistry." The references are missing. Please consider to revise. The statement is hard to follow

*Response*: we revised this sentence to :

*"Previous studies of pollution in Beijing have shown that the interactions of physical conditions, such as the development of temperature inversion in the atmosphere, and chemical processes, e.g., formation of secondary pollutants, such as aerosol particles and ozone that modulate the most severe air quality events."*

*Comment*: Lines 280-283, " AIRLESS aimed to advance air quality and health research in China by bringing together two fields of research that have made rapid advancements in recent years: measurements of a wide range of pulmonary and cardiovascular biomarkers in a panel study and personal monitoring of multiple air pollutants with high spatio-temporal resolution by sensor technology" In China or In Beijing and Neighboring Provinces? Why are the two sites' measurements helpful for the targets?

*Response*: All work is done at Beijing. We recognize that we could have been more specific. We have revised this sentence as below. We have identified the reasons why the site measurements are useful for AIRLESS (see below).

*" AIRLESS aimed to advance air quality and health research in Beijing by bringing together two fields of research that have made rapid advancements in recent years: measurements of a wide range of pulmonary and cardiovascular biomarkers in a panel study and personal monitoring of multiple air pollutants with high spatio-temporal resolution by sensor technology"*

*"AIRLESS is also benefiting from the use of an extensive range of pollution metrics and source apportionment results collected from in the Themes 1 and 2 projects. "*

**Reviewer 2:**

**Comment 1**: However, to be qualified as an ACP research article, the authors need to provide more sciences in the manuscript, as suggested below.

**Response**: This is not a research article but an introduction paper. Please see the general response above.

*Comment 2*: There have been many field campaigns, e.g., CAREBeijing, organized in the past 10 years targeted on investigating the air pollution and its health impact in Beijing. Compared with all those previous studies, what is unique about the current project? What are the scientific challenges this project aims to solve?

*Response*: Please see response to comment 1 by reviewer 1. APHH-Beijing is unique in that it is an integrated programme quantifying emissions of air pollutants using bottom-up, tower-based flux and satellite measurements, apportioning sources of particulate matter by multiple receptor models and hybrid receptor-chemical transport models, understanding atmospheric processes leading to pollution events via a coordinated measurements of physics and chemistry, and quantifying the health effects to individuals through personal exposure and health indicator / novel metabolomics measurements. APHH-Beijing integrates strengths in atmospheric sciences the UK with emerging research capabilities in China. In the newly added summary, we also highlighted the novel aspects of APHH-Beijing programme. Please also see response to reviewer comment 1.

*Comment 3*: There are four research themes presented: sources and emissions, atmospheric processes, health effects, and solutions. The last two only appeared in the very first part of the manuscript, no scientific output can be found later on. To make this manuscript completer and more consistent, primary results related to the health impacts need to be given.

*Response*: This is an introductory paper designed to set the scene rather than to give the conclusions of the study. Consequently, we feel that it would be inappropriate to report results from the health impacts studies which have yet to be subjected to peer review of their findings. It is our intention to write an overview paper towards the end of the APHH-Beijing programme where we will summerize the outcomes of the whole programme.

*Comment 4:* One focus of this manuscript is the overview of two joint field campaigns. Indeed, there are lots of discussions regarding the site information and type of instruments, but these discussions are not necessarily useful, as any future publications related to these two campaigns would have to give similar descriptions in their methods section anyway. Instead, this manuscript could be a nice platform for a detailed instrument calibration and comparison, data analysis and uncertainty quantification, and so on.

**Response**: One of the reasons why we have provided detailed site information is exactly to avoid every single paper to have a long paragraph describing exactly the same information. All further papers can refer to this paper for site information.

The list of instruments are very important part of the introduction paper as this gives readers an overview (big picture) of what is being measured within APHH-Beijing and see immediately if they might be able to find a particular type of data that they are interested in. Most single projects will make some particular type of measurements, but the APHH-Beijing programme made many complementary measurements.

We feel that instrument calibration is a routine work by each research group and this is too detailed for an introduction paper. Instead, such information should go to individual papers.

We take intercomparison extremely seriously within APHH-Beijing, particularly for those species that are still hard to measure, such as HONO. It is our intention that such highly specialized subjects will be published in individual papers rather than in this introduction paper. For example, an intercomparison HONO dataset has been generated for the whole APHH-Beijing programme to use, including both modelling and measurement scientists.

Data analysis and uncertainty quantification are highly complex in particularly when we are considering hundreds of species are being made. It is impractical to include such detailed information within an introduction paper.

**Comment 5**: The last two sections describe the air quality, e.g., the average concentrations and diurnal patterns of common air pollutants like NOx, O3, PM2.5, and etc., during the two field campaigns. As the authors highlighted earlier that regional modeling is an essential part of the campaigns, a modeling vs. observation comparison in terms of temporal profiles of these common pollutants need to be provided.
**Response**: We highly value this constructive comment and have added section 7 to address this comment. We also added two paragraphs:

"*Air quality modelling is a key aspect of APHH-Beijing. This involves multiple models from regional (e.g., WRF-Chem, UKCA, NAQPMS) to urban (e.g., CMAQ) and to street scales (ADMS). This section aims to provide an example comparison of model simulated pollutant concentrations against APHH-Beijing measurements made at IAP (Figure 16) to demonstrate model capabilities. Specific modelling work will be published in the special issue.*

*Figure 16 shows that the magnitude and variation of wintertime PM2.5 concentrations are reproduced very well by NAQPMS during November, although there are some weakness in capturing the highest PM2.5 levels during the haze events at the end of November and start of December. This is partly due to the representation of local meteorological features during this period, and PM2.5 concentrations during the major haze episode on 4 December are much more similar to those measured at Pinggu than at IAP (see Figure 2). The diurnal variations in O3 
[revised manuscript text omitted]
. CAREBEIJING was initiated and organized by Professor Tong Zhu of Peking University, with participation of hundreds of scientist and students from China, USA, Germany, Italy, Japan, and South Korea. The field campaigns were conducted in the summer of 2006, 2007, and 2008, with the objectives to learn the environmental conditions of the region, to identify the processes (transport and transformation) that lead to the impact of the surrounding area on air quality in Beijing, to quantify the impact of the surrounding area on air quality in Beijing, and to formulate policy suggestion for the air quality attainment during the 2008 Beijing Olympic Games. Other major research programmes, initiated since early 2000, aimed to provide scientific basis to deliver air pollution mitigation measures for ensuring a good air quality during the Olympics Games. Measures developed as a result of these programmes successfully reduced the air pollution during the Olympics Games, and provided valuable examples for air pollution control policy-making in other cities (Wang et al., 2010). CARE-BEIJING latter on was extended to CAREBEIJING-NCP (Campaigns of Air Pollution Research in Megacity Beijing and North China Plain), where field campaigns were carried out in the summer of 2013

and 2014 to investigate the transport and transformation processes of air pollutants in megacity Beijing and North China Plain. The results of CAREBEIJING and CAREBEIJIN-NCP have been published in three special issues of Atmospheric Chemistry and Physics (https://www.atmos-chem-phys.net/special_issue198.html) and Journal of Geophysical Research-Atmospheres (https://agupubs.onlinelibrary.wiley.com/doi/toc/10.1002/(ISSN)2169-8996.CARBS1). These large research programmes and numerous discovery science projects significantly enhanced our understanding on the emission, sources and processes of air pollutants in Beijing (Chan and Yao, 2008; Zhu et al., 2012). 
[revised manuscript text omitted]

During the summer campaign (Figure 12b), the most frequent CTs were 5, 8, 6, 7 (34, 32, 12, 11 % of the time, respectively). CT 8, which did not occur during the winter campaign period, is (like CT 6) associated with the summer monsoon advecting moist, warm air from the South and Southeast (Figure 11). Synoptic flow from the Northwest ( were CT 1 and 4) is relatively rare (7 and 4 %, respectively) during spring and summer (Mar-Aug) as  winds start to turn over the Yellow Sea, weakening the NW flow over Beijing.

In comparison to the field campaigns,  the CT frequencies range from 7.2% (CT 2, 10) to 12.9% (CT 8) during the period 1988-2017 with clear seasonal variations in their occurrence (Figure 13).

**6.2 Synoptic circulation and Air Quality**

The 11 CTs (Section 6.1) are clearly associated with distinct air quality conditions based on analysis of hourly air quality data for 2013-2017 at one of the national urban air quality stations (G4, Olympic Park, Figures 1 and 12). Relatively lower $PM_{2.5}$ concentrations occur (Figure  13b) under NE flow conditions (CTs 1-5), and higher concentrations during southerly flow (CTs 6-8, 10). The highest $PM_{2.5}$ concentrations occur during the heating season when regional flow shows stagnation (CT 9, 11). All haze events during the winter campaign (Figure 12) are dominated by those CTs although CTs with NE flow conditions occurred for short periods within the haze events (e.g. 18/11/2016, 04/12/2-16). Ozone levels are highest during CTs 5-8 (Figure  13c) as these predominate during spring and summer (Figure  13d).

Similarly, the average mixed layer height observed at IAP (Table 1) varies with season and CT type (Figure  13a). In the Oct 2016 – Sept 2017 period (Figure  13e), the relative frequency of CTs differs slightly from the long-term climatology (Figure  13d). In December 2016, clear air advection from the NE (CTs 1-3) was less frequent than in the 30-y climatology. However, stagnation with a weak southerly component (CTs 9 and 11) was more frequent (Figure  13f), thus favouring haze with a large positive (40%) $PM_{2.5}$ anomaly (Figure  14g, cf. 5 y average, 2013-2017). In June 2017, south-north contrasts in geopotential were apparently reduced so CT 6 was 24% less frequent, while CTs 4, 7, and 8 were more frequent. This had minimal effect of $PM_{2.5}$ The slight relative increase in $O_3$ (by 9.5%, Figure  13g) during June and January might be explained by  cloud cover differences, which is being investigated.

**6.3 Meteorological Conditions During the Field Campaigns**

To assess how local-scale flow related to ERA-Interim fields (section 6.1), the link between the coarse gridded data and tower-based sonic anemometer observations is explored based on wind roses (Figure 14). The 30 y climatology (Figure 13a, d) confirms the clear seasonality in wind direction affecting the occurrence of CTs discussed (Sect. 0), i.e. during winter intensive campaign period (5 November – 10 December) north-easterly flow clearly dominates while southerly wind directions are most common during the summer campaign period (15 May – 22 June). The wind roses for winter 2016 and summer

2017 (Figure 14b, e) are slightly nosier, however, indicating similar tendencies as the climatology. The
general large-scale patterns are consistent with the in-situ wind measurements (Figure 14c, f). However,
a slight diversion towards northerly and south-westerly flow and lower wind speeds occurred in winter
and summer (Figures 14c and f), respectively, when compared to the larger scale data (Figures 14b and
d).  In addition, south-westerly flows were more frequent in winter 2016 (Figures 14b and c) than the 30
year average climatology (Figure 14a), which had the potential to bring more polluted air in the upwind
Hebei province to the observation sites in Beijing.

At 102 m, the flow is consistent with northerlies and north-westerlies in the winter campaign and
dominantly southerly and easterlies during the summer campaign (Figure 15). The measured hourly mean
wind speed, temperature and relative humidity were 3.1 m s$^{-1}$, 8.3 $^{\circ}$C and 43.8 % in winter, and 3.6 m s,
25 $^{\circ}$C and 46.7 % in summer, respectively. Typical diurnal patterns were observed with higher wind speed
and temperature during the day and RH at night. During the winter haze events the 120 m wind speed
were low (an average of 1.8 m s$^{-1}$) and mainly from the south-west direction (Figures 15 and 2).

**6.47.   PRELIMINARY AIR QUALITY MODELLING AND POLLUTION CLIMATOLOGY OF THE CAMPAIGN PERIODS**

Air quality modelling is a key aspect of APHH-Beijing. This involves multiple models from regional
(e.g., WRF-Chem, UKCA, NAQPMS) to urban (e.g., CMAQ, NAQPMS) and to street scales (ADMS).
This section aims to provide an example comparisons of model simulated pollutant concentrations
against APHH-Beijing measurements made at IAP (Figure 16) to demonstrate model capabilities.
Specific modelling work will be published in the special issue later.

Figure 16 shows that the magnitude and variation of wintertime PM$_{2.5}$ concentrations are reproduced
very well by NAQPMS during November, although there are some weakness in capturing the highest
PM$_{2.5}$ levels during the haze events at the end of November and start of December. This is partly due to
the representation of local meteorological features during this period, and PM$_{2.5}$ concentrations during
the major haze episode on 4 December are much more similar to those measured at Pinggu than at IAP

[revised manuscript text omitted]

---

## Author Response (AR2)

Journal: ACP

10

- 2 MS No.: acp-2018-922
- 3 Title: Introduction to Special Issue In-depth study of air pollution sources and processes within Beijing and
- 4 its surrounding region (APHH-Beijing)
- 5 Author(s): Zongbo Shi et al.
- 6 Iteration: Major Revision
- 7 Special Issue: In-depth study of air pollution sources and processes within Beijing and its surrounding region
- 8 (APHH-Beijing) (ACP/AMT inter-journal SI)
  - **RESPONSE TO REVIEWER**

*General Response*: We thank the reviewer for these very helpful comments. We thank the reviewer for supporting the need of an introduction paper to this special issue and for agreeing that we have responded well the previous round of reviews.

Comment 1: Serious efforts are deemed necessary to improve the writing quality, logical flow, and clarity. The current version contains many parts that are vague, hard to follow, and abundant of grammatical errors. The manuscript also sometimes reads like a compilation of texts from multiple authors without a streamlined integration. Several issues are pointed out below in this review but the list is far from complete. A careful proofreading and English editorial job would be necessary.

**Response**: We have carefully re-written almost all parts of the papers to improve the writing quality, logical flow and for clarity. In order to improve the flow, the review of meteorological conditions is brought forward after the description of the campaign sites and instrumentation and before the brief review of measurement data.

Comment 2: The article is also on the long side with details that appear to be redundant and somewhat distracting.

**Response**: In the absence of more information, we were unable to identify the specific problems and make an informed revision. We did, however, look at the whole paper, revise the paper substantially, and removed information that we felt not critical to the paper (e.g., Fig. 1c) and moved five figures to the supplement.

Comment 3: The scientific values and innovative aspects of the project should be emphasized more. *Response*: We thank the reviewer for this comment and now emphasized the novelty of the APHH-Beijing project in section 3.2.

Page 10 Line 338: "Together, this interdisciplinary APHH-Beijing programme delivers key scientific values and innovation, including:

(1) Validation of the bottom-up emission inventories by using novel eddy covariance emission flux observations from the IAP meteorological tower, integrated with satellite retrievals and numerical modelling, (2) Improvement in understanding of air pollution processes through comprehensive observations of atmospheric gaseous and aerosol species integrated with atmospheric physics measurements, and (3) Identification of the sources of air pollution that cause adverse human health effects by carrying out novel cardiovascular health indicator measurements, integrated with personal exposure and fixed station source apportionment studies."

Comment 4: Page 2-3, the revised abstract shows no improvement and remains ineffective. The revisions and rearrangement of the textst are sometimes confusing. For example, information on "the two campaigns" should be mentioned before the sentence on line 80. The added discussions on pollution concentrations do not fit well. The abstract should be rewrittend.

**Response**: In the absence of more information, we were unable to identify the problems with the abstract. Thus, we have gone back to past published introduction papers to special issues and attempted to re-write this abstract following the styles in those papers. We put much emphasise in introducing the APHH-Beijing programme and attempted to improve the flow of information. We also moved the detailed information on air quality and meteorology during the campaigns to the end of section 5 (i.e., section 5.5. Summary of air quality during the campaigns).

Comment 5: Line 125, it is strange to claim Mexico City as a developed megacity while Beijing a developing one. *Response*: Mexico City has been taken out of the list of developed megacities.

Comment 6: Line 252, is AIRPOLL-Beijing part of APHH-Beijing? Multiple acroynmed studies, sucha as AIRPRO, AIRELESS, APIC-ESTEE, INHANCE, etc, are discussed, but their relationships to APHH are unclear. It could be helpful to add a paragraph earlier in the manuscript to introduce these studies and explain their relationships with APHH.

**Response**: APHH-Beijing is a large research programme including four research themes delivered by five research projects. Theme 1 and 2 are addressed by AIRPOLL-Beijing and AIRPRO respectively. Theme 3 is addressed by two projects - APIC-ESTEE and AIRLESS and Theme 4 by INHANCE.

We have now added information at the beginning of Section 3 (Line 201 1):

"The APHH-Beijing programme has four themes to address the specific objectives outlined in Section 2, and is delivered through five inter-related research projects. :

- Theme 1 - Sources and emissions: delivered by the AIRPOLL-Beijing (Source and Emissions of Air Pollutants in Beijing) project;

- Theme 2 – Atmospheric processes: delivered by the AIRPRO (The integrated Study of AIR Pollution PROcesses in Beijing) project;

- Theme 3 – Health effects: delivered by the AIRLESS (Effects of AIR pollution on cardiopuLmonary disEaSe in urban and peri-urban reSidents in Beijing) and APIC-ESTEE (Air Pollution Impacts on Cardiopulmonary Disease in Beijing: An integrated study of Exposure Science, Toxicogenomics and Environmental Epidemiology) projects;

- Theme 4: Solutions: delivered by the INHANCE (Integrated assessment of the emission-health-socioeconomics nexus and air pollution mitigation solutions and interventions in Beijing) project."

**Other points -**

**RESPONSE:** All of these have been addressed in editing the text into better English.

**Introduction to Special Issue - In-depth study of air pollution sources 14 and processes within Beijing and its surrounding region 15 (APHH-Beijing) 16**

Zongbo Shi1,2\*, Tuan Vu1, Simone Kotthaus3,4, Roy M. Harrison1†, Sue Grimmond3, Siyao Yue5, Tong 18 Zhu6, James Lee7,8, Yiqun Han6,9, Matthias Demuzere10, Rachel E Dunmore7, Lujie Ren2,5, Di Liu1, Yuanlin Wang5,11, Oliver Wild11, James Allan12,13, Joe Acton11, Janet Barlow3, Benjamin Barratt9, 19 20 David Beddows1, William J, Bloss1, Giulia Calzolai14, David Carruthers15, David C Carslaw7,16, 21 Queenie Chan9, Lia Chatzidiakou17, Yang Chen18, Leigh Crilley1, Hugh Coe12, Tie Dai5, Ruth 22 Doherty19, Fengkui Duan20, Pingqing Fu2,5, Baozhu Ge5, Maofa Ge21, Daobo Guan22, Jacqueline F. 23 Hamilton7, Kebin He20, Mathew Heal19, Dwayne Heard23, C Nicholas Hewitt11, Michael Hollaway11, 24 Min Hu6, Dongsheng Ji5, Xujiang Jiang20, Rod Jones17, Markus Kalberer17,a, Frank J Kelly9, Louisa Kramer1, Ben Langford24, Chun Lin19, Alastair C Lewis7, Jie Li5, Weijun Li25, Huan Liu20, Junfeng 25 26 Liu26, Miranda Loh27, Keding Lu6, Franco Lucarelli14, Graham Mann28, Gordon McFiggans12, Mark R. 27 Miller29, Graham Mills30, Paul Monk31, Eiko Nemitz24, Fionna O'Connor32, Bin Ouyang11,17, Paul I. 28 Palmer19, Carl Percival12,b, Olalekan Popoola17, Claire Reeves30, Andrew R Rickard7,8, Longyi Shao33, Guangyu Shi5, Dominick Spracklen28, David Stevenson19, Yele Sun5, Zhiwei Sun34, Shu Tao26, 29 30 Shengrui Tong21, Qingqing Wang5, Wenhua Wang33, Xinming Wang35, Xuejun Wang26, Zifang Wang5, Liangfang Wei5, Lisa Whalley23, Xuefang Wu1, Zhijun Wu6, Pinhua Xie36, Fumo 31

Yang37, Qiang Zhang38, Yanli Zhang35, Yuanhang Zhang6, Mei Zheng6 33

[revised manuscript text omitted]

**153**

Many research programmes were have been initiated in Beijing to study the air pollution processes since the late 1990s. Earlier research programmes (e.g., early 2000) focused on primary emissions of SO2, NO2, CO, PM10, volatile organic compounds, and then subsequently secondary pollutants such as ground-level ozone and secondary fine particles. Theisse researches contributed to the development of air pollution mitigation strategies introduced by the Beijing Municipal government.

The Beijing Olympic Games (2008) offered additional incentives to improve air quality and this led to 161 the funding of CAREBEIJING (Campaigns of Air Pollution Research in Megacity Beijing and 162 Surrounding Region) and other major programmes. CAREBEIJING was initiated and organized by 163 Professor Tong Zhu of Peking University, with participation of hundreds of scientist and students from 164 China, USA, Germany, Italy, Japan, and South Korea. The field campaigns were conducted in the summer 165 of 2006, 2007, and 2008, with the objectives to learn the environmental conditions of the region, to identify and quantify the processes (transport and transformation) that lead to the impact of the 167 surrounding area on air quality in Beijing, to quantify the impact of the surrounding area on air quality in Beijing, and to formulate policy suggestions for the air quality attainment improvement during the 2008 168 169 Beijing Olympic Games. Other major research programmes, initiated since early 2000, aimed to provide 170 scientific basis to deliver air pollution mitigation measures for ensuring a good air quality during the 171 Olympics Games. Measures developed as a result of these this and other programmes successfully 172 reduced the air pollutionimproved air quality during the Olympics Games, and provided valuable 173 examples for developing air pollution control policy-making in other cities (Wang et al., 2010). CARE-174 BEIJING latter on was later extended to CAREBEIJING-NCP (Campaigns of Air Pollution Research in 175 Megacity Beijing and North China Plain), where in which field campaigns were carried out in the summer 176 of 2013 and 2014 to investigate the transport and transformation processes of air pollutants in megacity the Beijing megacity and North China Plain. The results of CAREBEIJING and CAREBEIJING-NCP 177 178 have been published in three special issues of Atmospheric Chemistry and Physics (https://www.atmos-Journal 179 chem-phys.net/special\_issue198.html) and of Geophysical **Research-Atmospheres** 180 (https://agupubs.onlinelibrary.wiley.com/doi/toc/10.1002/(ISSN)2169-8996.CARBS1). These large research programmes and numerous discovery science projects significantly enhanced our understanding 181 on the emission, sources and processes of air pollutants in Beijing (Chan and Yao, 2008; Zhu et al., 2012). 182 However, our understanding of sources and emissions of key air pollutants such as PM2.5 and ozone and 183 184 the role of the interactions between physical and chemical processes in the formation-development of 185 pollution events in the Beijing megacities is still far from being accurate or complete. In addition, none of the abovementioned large programmes are have been directly linked to health effect studies. 186

The Aadverse health effects of air pollution is provide one of the key motivations to control air pollution. Research has shown that air pollution is one of the leading causes of the disease burden in China (GBD 188 189 MAPS Working Group, 2016). Especially, particulate pollution, the leading cause of severe air pollution 190 events in China, has a significant impact on human health and is associated with high mortality (Zhang 191 et al., 2017a), with a considerable proportion of this related to cardiorespiratory diseases (namely stroke, 192 ischemic heart disease, and chronic obstructive pulmonary disease) (Yang et al., 2013; Lozano et al., 193 2013). Despite this increasing evidence base, the adverse health impact of air pollution remains a complex 194 issue. For instance, the risk assessment of disease burden due to air pollution in China has relied largely 195 on the studies undertaken in Europe and North America, which likely over simplifies estimates may be subject to error due to the difference of race, life style, and air pollution settings (Lim et al., 2012). The 196 197 marked change in air pollution sources and composition between the heating and non-heating seasons, and the differences between urban and rural areas may all lead to different biological responses in local 198 199 residents populations. However, to date, such comparative investigations are largely lacking. A further limitation of such work is the lack of accurate personal exposure estimates which are crucial in high quality health studies. This may be especially true when considering household air pollution (both indoors and outdoors) from traditional biomass and coal stoves which may not be easily captured by ambient locatedtypical outdoor monitoring instruments (Linn et al., 2001; Brook et al., 2002). Thus, To address current uncertainties and challenges it is essential to improve understanding of the health impact of air pollution worldwidein China remains a major challenge, and to develop mitigation measures with limited resources onwhich reduce pressures upon health services.

To address these issues, the UK Natural Environment Research Council (NERC), in partnership with the 207 208 National Science Foundation of China (NSFC), UK Medical Research Council (MRC) and UK-China Innovation Newton Fund funded a major joint research programme – Atmospheric Pollution and Human 209 210 Health in a Chinese Megacity (APHH-Beijing). The APHH-Beijing is an integrated research programme, 211 incorporating the capabilities and strengths of the UK and Chinesea science communities and which 212 is taking a multi-disciplinary approach to investigating the sources, processes and health effects of air pollution in the Beijing megacity(1) sources and emissions of urban atmospheric pollutants; (2) processes 213 214 affecting urban atmospheric pollution; and (3) the exposure and impacts of air pollution on human health. The new scientific understanding from these three themes underpins the development of interventions 215 216 and solutions to improve air quality and reduce health impacts.

This special issue "In-depth study of air pollution sources and processes within Beijing and its surrounding region (APHH-Beijing)" documents the research outcomes of this APHH-Beijing programme, in particular the atmospheric measurement and modelling aspects.

This introduction paper describes the motivation and background of the APHH-Beijing programme, and 222 presents some of the background air quality and meteorologicaly observations, particularly during the 223 two intensive field campaigns. These campaigns form one of the core research activities within APHH-224 Beijing integrating the different themes / projects. We did not intend todo not present the key scientific 225 results of APHH-Beijing here-in this introduction (not an overview) paper as much of the research 226 activityies are still ongoing and unpublished. Such information is more suitable to go to an overview paper insteadKey 
[revised manuscript text omitted]

**324 3.1.2 Theme 2: Atmospheric processes**

The aims of the AIRPRO aims project aims are to study the basic fundamental chemical and physical 326 processes controlling gas and aerosol-particle pollution, localised meteorological dynamics, and the links 327 between them within Beijing's atmosphere. Once released to air, atmospheric processing controls how 328 pollutants are subsequently deposited, transformed into secondary pollutants such as O3 and particulate 329 matter (PM) or transported away from or within the wider Beijing urban area. Previous studies of pollution 330 in Beijing have shown that the interactions of physical conditions, such as the development of temperature 331 inversions in the atmosphere, and chemical processes, e.g., formation of secondary pollutants, such as aerosol particles and ozone 
[revised manuscript text omitted]
 took place from  $105^{\text{th}}$  November to  $10^{\text{th}}$  December 2016 and  $15^{\text{th}}-20^{\text{th}}$ May to  $22^{\text{nd}}$  June 2017. The campaigns were carried out at both urban and rural sites.

**446 4.1 Site Information**

The winter campaign hads two main sites. The urban site (39.97N, 116.38 E) is located in the Tower Section of the Institute of Atmospheric Physics (IAP), Chinese Academy of Sciences; where i.e. at the 448 449 325 m meteorological tower is located. The site, between the fourth and third North ring roads of Beijing (Figure 1), is in a residential area. Typical of central Beijing, there are various roads nearby. To the south, 450 451 north and west there are roads about 150 m away. On site there are 2 to 3 floor buildings to the south, and 452 the east and west of the tower surroundeding by small trees and grasses. There is a canal right to the north 453 of the site. Further to the west is a park covered mainly by conifer pine trees (Yuan Dynasty Wall Heritage 454 Park).

The rural Pinggu site in Xibaidian village (40.17N, 117.05 E) in north-eastern Beijing, was collocated with the AIRLESS project cohort. Xibaidian village is about 4 km northwest of Pinggutown centre, and about 60 km from IAP. There are many-several similar small villages nearby. The monitoring station and the clinic used an unoccupied house at the north end of the village away from significant local combustion sources. A two-lane road is about 300 m north to-of the site. With no centralised heating infrastructure available, to the local villages' residents mainly use coal and biomass for heating and cooking in individual homes.

In the summer, an additional site was operated in Gucheng (39.2N 115.7E), Dingxing County, Hebei Province. This site, about 120 km to the southwest of central Beijing, is on one of the main highly pollutant transport pathways from Hebei province to Beijing via from the southwest passage. The site used is in a meteorological observatory surrounded by in a 
[revised manuscript text omitted]

\$57

**5.1 Winter**

5. <u>Air Quality During the Field CampaignsAIR QUALITY DURING THE FIELD 567 CAMPAIGNS

During the winter campaign, the daily average concentration of PM2.5 at IAP using Partisol gravimetric 569 measurements was 91.2 µg m-3 from the Partisol gravimetric measurements (Table 4) and 94.0 µg m-3 570 from an online FDMS (Filter dynamic measurement system)-measurements. The maximum hourly PM2.5 \$71 concentration was 438 µg m-3 (Figure 275 which shows the haze events listed in Table 5). PM2.5 572 concentrations significantly exceeded the both the daily air quality limit of China (75 µg m-3) and WHO \$73 (25 µg m-3). During the whole winter campaign period, nearly 50% of the hours had PM2.5 mass 574 concentration higher than 75  $\mu$ g m-3 (Figure 275). The Oonline PM10 concentration observed at the \$75 Olympic Park national air quality monitoring station was up to 560  $\mu$ g m-3 during the campaign with an 576 average of 130.6  $\mu$ g m-3. Average concentrations of NO2, O3, SO2 and CO were 69.7 ± 33.3, 16.4 ± 17.0 577 and  $14.9 \pm 11.1 \ \mu g \ m^{-3}$  and  $1.53 \pm 1.02 \ m g \ 
[revised manuscript text omitted]

- 659
- Average concentrations of air pollutants (PM2.5, PM10, NO2, CO, SO2 and O3) at IAP and Pinggu during the two field campaigns were similar to long term averages for these times of year at the 12 national air quality monitoring sites for 2013 2017 (Table 4). In general, PM2.5 mass concentrations are similar at all the urban sites including IAP but which are higher than at the suburban and rural background national monitoring sites (Ming Tombs, G2, G9 and G10, ) (Figure S4-914). The Pinggu rural site in this study, has relatively high PM2.5 pollution in during the winter campaign but has the lowest concentrations during the summer campaign. This suggests that local anthropogenic sources have a major impact on PM2.5 at this site during the winter campaigns. Source apportionment results, notably high time resolution data are
being used to explore this.

[revised manuscript text omitted]

During the summer campaign (Figure 12b), the most frequent CTs were 5, 8, 6, 7 (34, 32, 12, 11 % of the time, respectively). CT 8, which did not occur during the winter campaign period, is (like CT 6) associated with the summer monsoon advecting moist, warm air from the South and Southeast (Figure 11). Synoptic flow from the Northwest (CT 1 and 4) is relatively rare (7 and 4 %, respectively) during spring and summer (Mar Aug) as winds start to turn over the Yellow Sea, weakening the NW flow over Beijing.

In comparison to the field campaigns, the CT frequencies range from 7.2% (CT 2, 10) to 12.9% (CT 8)
 during the period 1988-2017 with clear seasonal variations in their occurrence (Figure 13).

**728 6.25.4 Synoptic Ceirculation and Air Quality**

The average mixed layer height observed at IAP varies with season and CT type (Figure 10a). Lower mixed layer height is usually linked to air pollution events. The 11 CTs (Section 6.14.3) are clearly associated with distinct air quality conditions based on analysis of hourly air quality data for 2013-2017 at one of the national urban air quality stations (G114, Olympic Park, Figure 1s 1 and 123).-Relatively 733 low wind speeds (Figure 7) of CT 7 may contribute to the long haze event from 15/11/2016 to 19/11/2016, (Fig. 5). Most haze events during the winter campaign are cleared out by fresh air masses being advected 734 from the North in CTs 3 or 5 (Figure 3), which is also marked by the increase in wind speed observed 735 736 (Figure \$16). Relatively lower PM2.5 concentrations occurred (Figure 410b) under NE flow conditions 737 (CTs 1-5), and higher concentrations during southerly flow (CTs 6-8, 10). The highest PM2.5 concentrations occur during the heating season when regional flow shows showed stagnation (CT 9, 11). 738 739 All haze events during the winter campaign (Figures 3&1235) are dominated by those CTs although CTs 740 with NE flow conditions occurred for short periods within the haze events (e.g. 18/11/2016, 04/12/2-16). 741 Ozone levels are highest during CTs 5-8 (Figure 13c104c), which as these predominate during spring 742 and summer (Figure 410d).

Similarly, the average mixed layer height observed at IAP (Table 1) varies with season and CT type 745 (Figure 4a). InDuring the Oct 2016 Sept 2017 period (Figure 410e), the relative frequency of CTs differs slightly from the long term climatology (Figure 410d). In December 2016, clearn air advection from the 746 NE (CTs 1-3) was less frequent than in the 30-y climatology. However, while stagnation with a weak 747 southerly component (CTs 9 and 11) was more frequentoccurred more often (Figure 410f), thus favouring 748 749 haze with a large positive (4032%) PM2.5 anomaly (Figure 510g, winter campaign period compared to cf. the 5 y average, for 2013-2017). In June 2017, south-north contrasts in geopotential were apparently 750 reduced so CT 6 was 24% less frequent, while CTs 4, 7, and 8 were more frequent. This had minimal 751 effect of PM2.5. The slight relative increase in O3 (by 9.5%, Figure 410g) during June (9%) and January 752 (12%) might be explained by cloud cover differences, which is being investigated. 753

[revised manuscript text omitted]

76. Preliminary Air Quality Modelling and Pollution Climatology of the Campaign Periods 801 Air quality modelling is a key component of the APHH-Beijing programme. A range of models have been applied that span global (UKCA, GEOS-Chem), regional (WRF-Chem, CMAQ, NAQPMS) and 802 803 urban to street scales (ADMS). This section provides an example of the comparison between model 804 simulated pollutant concentrations and APHH-Beijing observations made at IAP to demonstrate model capabilities. Results from specific modelling studies will be published separately. 805 806 Air quality modelling is a key aspect of APHH-Beijing. This involves multiple models from regional 807 (e.g., WRF-Chem, UKCA, NAQPMS) to urban (e.g., CMAQ, NAQPMS) and to street scales (ADMS). 808 This section aims to provide an example comparison of model simulated pollutant concentrations against APHH-Beijing measurements made at IAP (Figure 16) to demonstrate model capabilities. 809 810 Specific modelling work will be published in the special issue later. 811

Figure 16-11 shows that the magnitude and variation of wintertime PM2.5 concentrations are reasonably reproduced by NAQPMS during November the winter campaign, although there are some weakness in 813 capturing the highest PM2.5 levels during the haze events at the end of November and start of December. 814 815 This is partly due to the representation of local meteorological features during this period, which bring these episodes to an end 6-12 hours early.and PM2.5 concentrations during the major haze episode on 4 816 December are much more similar to those measured at Pinggu than at IAP (see Figure 27). The diurnal 817 variations in O3 during the summertime are reproduced relatively well by UKCA, which captures the 818 819 rapid daytime formation of O3 and strong nighttime removal. The very highest levels of daytime O3 are 820 underestimated with the model, particularly during the episode at the end of May. However, there is a 821 strong local contribution to this as evident from the lower concentrations measured at Pinggu (Figure 8), 822 and these local differences are not fully resolved with the model. Despite this, the day-on-day build-up 823 of daytime O3 during the periods of 22-27 May and 11-16 June is captured-well, and demonstrates that 824 the model reproduces the synoptic drivers of local O3 formation well.

We also investigated how representative the campaign periods were of the selected seasons in Beijing by 827 comparing pollutant levels with those from the same period each year over the 2013-2017 period. The NAQPMS model was run for the full 5-year period driven by NCEP meteorology and using temporally 828 829 varying emissions for a single year that is broadly representative of 2013 conditions. The same emissions were used each year so that the meteorological contribution to pollutant levels could be assessed. This 830 provides important information that cannot be obtained from the monitoring data (as emission varies year 831 by year). The frequency distribution of PM2.5 over each campaign period for each year is shown in Figure 832 12. Winter 2016 was broadly typical of the 5-year period, with similar characteristics to winter 2014, but 833

both years show higher  $PM_{2.5}$  under the same emissions than in 2013 or 2017. In addition, winter 2015 had substantially less favourable conditions for air quality, and more stagnant conditions led to three extended pollution episodes over the period with  $PM_{2.5}$  exceeding 200 µg m-3. In contrast, the summer period in 2017 was cleaner than average, with  $PM_{2.5}$  levels very similar to 2015, and about 25% less than in 2013, 2014 or 2016. These results are broadly consistent with those based on synoptic weather analyses (section 5.4) as well as by Vu et al. (2019).

On basis of Based on NAQPMS modelling, we also investigated how representative the campaign periods 840 were of the selected seasons in Beijing by comparing pollutant levels with those from the same period 841 each year over the 2013-2017 period. The NAOPMS model was run for the full 5-year period driven by 842 NCEP meteorology and using temporally varying emissions for a single year that is broadly representative 843 844 of 2017 conditions. Use of annually invariant emissions permits the effect of differing meteorology on pollutant levels to be assessed. The frequency distribution of PM2.5 for each campaign period for each 845 year is shown in Figure 17. PM2.5 in winter 2016 is very similar in characteristics to that in 2014, and 846 both years show 50% greater PM levels than in 2013 or 2017. However, pollutant levels are substantially 847 lower than in the same period in 2015, when three extended pollution episodes led to period-mean PM2.5 848 that was significantly larger. In contrast, the summer period in 2017 was relatively clean, with PM2.5 849 levels very similar to 2015, and about 25% less than in 2013, 2014 or 2016. 850

**852 87. SUMMARYSummary**

The APHH-Beijing is an integrated and multidisciplinary research programme conducted by leading UK 853 and Chinesea researchers to (1) quantify sources and emissions of urban atmospheric pollutants; (2) 854 elucidate processes affecting urban atmospheric pollution events; (3) estimate the personal exposure and 855 impacts of air pollution on human health, and (4) develop intervention strategies to improve air quality 856 857 and reduce health impacts in the Beijing megacity. This introduction paper outlines the motivation of the 858 APHH-Beijing programme as well as providinges the background air quality and meteorological conditions during the two intensive field campaigns that form the basis of data interpretation for the whole 859 programme, particularly campaign observations during the two intensive field campaigns as a core 860 research activity within the programme. 861

[revised manuscript text omitted]
., 18, 2018. 1421 1422 Zhu, T., Melamed, M. L., Parrish, D., Gauss, M., Klenner, L. G., Lawrence, M. G., Konare, A., Liousse, 1423 C.: WMO/IGAC Impacts of Megacities on Air Pollution and Climate, GAW Report No. 205; 205, 1424 Geneva : World Meteorological Organization Global Atmosphere Watch, 2012. 1425 https://www.wmo.int/pages/prog/arep/gaw/documents/GAW\_205\_DRAFT\_13\_SEPT.pdf (last accessed 1426 01/02/2019). 1427 1428 1429

[revised manuscript text omitted]

|             | OH reactivity           | OH reactivity                                                  | Leeds         | Stone et al. (2016)           |  |  |
|-------------|-------------------------|----------------------------------------------------------------|---------------|-------------------------------|--|--|
|             | Spectral radiometer     | Photolysis rates                                               | Leeds         | Bohn et al. (2016)            |  |  |
|             | Filter radiometer       | $J(O^1D)$                                                      | Leeds         | Bohn et al. (2016)            |  |  |
| 0           | Dew point               | Water vapour                                                   | Leeds         | Whalley et al. (2010)         |  |  |
| Container 2 | nygrometer              | XX7' 1 1 1' 4' 4                                               |               |                               |  |  |
|             | Davis met station       | Wind speed, direction, temp,
RH, pressure                   | Leeds         |                               |  |  |
|             | Voicele CL 21 ALC       | Cloud-base height, mixing                                      |               | Kotthaus and Grimmond         |  |  |
|             | Ceilometer + | height, attenuated backscatter profiles                        | Reading       | (2018a)                       |  |  |
|             | Personal air            | CO, NO, NO 2 , PM 1 , PM 10 , | a 1 11 |                               |  |  |
|             | monitors (PAMS)         | PM 2.5                                              | Cambridge     | Moore et al. (2016)           |  |  |
|             | MicroPEMs               | Personal PM exposure                                           | IOM           | Sloan et al. 2015             |  |  |
|             |                         |                                                                |               |                               |  |  |
|             | DC-GC-FID               | C2-C7 VOCs and oVOCs                                           | York          | Hopkins et al. (2011)         |  |  |
|             | GCxGC FID               | C6 - C13 VOCs and oVOCs                                        | York          | Dunmore et al. (2015)         |  |  |
|             | TEI 42i                 | NO                                                             | Birmingham    |                               |  |  |
| ŗr 2        | Teledyne CAPS           | $NO_2$                                                         | York          |                               |  |  |
| aine        | TEI 42c                 | Total NO y                                          | York          |                               |  |  |
| Cont        | TEI 49i                 | O 3                                                 | York          |                               |  |  |
| •           | TEI 43i                 | $SO_2$                                                         | York          |                               |  |  |
|             | Sensor box              | СО                                                             | York          | Smith et al. (2017)           |  |  |
|             | BBCEAS                  | HONO, NO 3 , N 2 O 5          | Cambridge     | Le Breton et al. (2014)       |  |  |
|             |                         |                                                                |               |                               |  |  |
|             | LOPAP                   | HONO                                                           | Birmingham    | Crilley et al. (2016)         |  |  |
| ŝ           | LIF HCHO                | НСНО                                                           | Leeds         | Cryer et al. 2016             |  |  |
| ner         | LOPAP                   | HONO                                                           | IC-CAS        | Zhang et al. (201 9 8) |  |  |
| ntai        | GC-MS                   | Organic nitrates                                               | East Anglia   | Mills et al. (2016)           |  |  |
| Co          | ROS online
analyser  | Reactive Oxygen Species                                        | Cambridge     | Wragg et al. (2016)           |  |  |
|             |                         |                                                                |               |                               |  |  |
| Con         | FAGE                    | OH (wave) x , HO 2                       | Peking        | Lu et al., 2012               |  |  |

|           | FAGE                        | OH (chem) x                            | Peking           | Tan et al., 2017        |  |  |
|-----------|-----------------------------|---------------------------------------------------|------------------|-------------------------|--|--|
|           | TEI 42i                     | NO                                                | Peking           | Tan et al., 2017        |  |  |
|           | Teledyne CAPS               | NO2                                               | Peking           |                         |  |  |
|           | TEI 42c with Moly converter | NO 2                                   | Peking           |                         |  |  |
|           | TEI 49i                     | O 3                                    | Peking           |                         |  |  |
|           | TEI                         | СО                                                | Peking           |                         |  |  |
|           | Spectral radiometer         | Photolysis rates                                  | Peking           |                         |  |  |
|           | GC-ECD                      | PAN                                               | Peking           | Zhang et al., 2011      |  |  |
|           | GC-MS                       | VOCs                                              | Peking           | Wang et al., 2015a      |  |  |
|           |                             |                                                   |                  |                         |  |  |
|           | H-TDMA/V-
TDMA           | Hygroscopicity/volatility                         | Peking           | Wu et al., 2013         |  |  |
| ·5 *      | SMPS+APS                    | Particle Number size distribution                 | Peking           | Wu et al., 2016         |  |  |
| Container | Particle size magnifier     | Size distribution of < 3nm particles              | Peking           | Vanhanen et al., 2011   |  |  |
| 0         | IGAC-IC                     | Water-soluble ions                                | Peking           | Yu et al. (2018)        |  |  |
|           | Xact                        | Metal                                             | Peking           | Yu et al. (2018)        |  |  |
|           | Sunset OC/EC                | EC/OC                                             | Peking           | Zhang et al. (2017b)    |  |  |
|           |                             |                                                   |                  |                         |  |  |
|           | IBBCEAS                     | HONO, NO 2                             | AIOFM            | Duan et al. (2018)      |  |  |
|           | CRDS                        | NO 3 and N 2 O 5 | AIOFM            | Li et al. (2018)        |  |  |
| er 6      | Nitrate Api-TOF-            |                                                   |                  |                         |  |  |
| tain      | CIMS                        | Organics, clusters (HOMs)                         | Birmingham       | Junninen et al. (2010)  |  |  |
| Con       | SMPS                        | Particle size distribution                        | Birmingham       | Shi et al. (1999)       |  |  |
|           | Particle size               | Size distribution of $< 3 \text{ nm}$             |                  |                         |  |  |
|           | magnifier                   | particles                                         | Birmingham       | Vanhanen et al. (2011)  |  |  |
|           |                             |                                                   | X 7 1     |                         |  |  |
|           | Fast $NO_x$                 | NO x fluxes                            | YOrk             | Vaughan et al. (2016)   |  |  |
| 7         | AL5002 CO
analyser       | CO fluxes                                         | York             | Gerbig et al. (1999)    |  |  |
| iner      | HR-TOF-AMS                  | Fluxes of PM1 non-refractory                      | CEH              | Nemitz et al. (2008)    |  |  |
| onta      |                             | (NR) species                                      |                  | 110111112 et al. (2008) |  |  |
| Ŭ         | SP2                         | BC fluxes                                         | Manchester       | Liu et al. (2017)       |  |  |
|           | PTR-TOF-MS                  | VOC fluxes                                        | GIG
Lancaster | Huang et al. (2016)     |  |  |

|             | SYFT-MS Voice
Ultra | VOC fluxes                                      | York       | Storer et al. (2014)           |  |
|-------------|----------------------------|-------------------------------------------------|------------|--------------------------------|--|
|             | SMPS3968-                  | Particle number size                            | BNU        | Du et al. (2017)               |  |
| Container 8 | H/V TDMA                   | Particle hyprosconicity                         | BNU        | Wang et al. (2017b)            |  |
|             | CCNC-100                   | CCN                                             | BNU        | Wang et al. (2017b)            |  |
|             | PAX (870nm)                | Extinction & absorption coefficient             | IAP        | Xie et al. (2018)              |  |
|             | Ammonia analyzer           | NH 3                                 | IAP        | Meng et al. (2018)             |  |
|             | Sunset OC/EC
analyzer   | Online OC/EC                                    | IAP        | Zhang et al. (2017b)           |  |
|             | Iodide FIGAERO-            | Particle and gas phase molar                    |            |                                |  |
| ır 9        | TOF-CIMS                   | molecule                                        | Manchester | Le Breton et al. (2018)        |  |
| Containen   | CPMA-SP2                   | Black carbon mass and mixing state              | Manchester | Liu et al. (2017)              |  |
| •           | Micro reactor              | oVOCs                                           | York       | Pang et al. (2014)             |  |
|             |                            |                                                 |            | M M (2010)                     |  |
|             | QCL NH 3        | Ammonia fluxes                                  | СЕН        | McManus et al. (2010)          |  |
|             | IRGA LiCOR-
| CO 2 / H 2 O flux         | СЕН        | McDermitt et al. (2011)        |  |
|             | DMT UHSAS                  | Size resolved particle flux (0.06-1 µm)         | СЕН        | Deventer et al. (2015)         |  |
| (00 m       | TSI APS3021                | Size-resolved particle flux (0.5-25 µm)         | СЕН        | Nemitz et al., (2002)          |  |
| Tower ~1    | TSI CPC3785                | Total particle number flux                      | СЕН        | Petäjä et al., (2006)          |  |
|             | ROFI                       | O 3 flux                             | СЕН        | Coyle et al., 2009             |  |
| T           | Sonic anemometer
R3-50  | Turbulence, sensible heat flux                  | СЕН        | Högström and Smedman
(2004) |  |
|             | WXT530 weather station     | T, P, RH, wind speed & direction, precipitation | СЕН        |                                |  |

|          | 2B O 3 analyser | O 3 concentration                                  | CEH        | Johnson et al. (2014) |  |  |
|----------|----------------------------|---------------------------------------------------------------|------------|-----------------------|--|--|
| m 03     | High-vol sampler           | PM 2.5 filter samples                              | IAP        |                       |  |  |
| r ~12    |                            |                                                               |            |                       |  |  |
| Гоње     | Anderson sampler           | Size-resolved PM samples                                      | IAP        |                       |  |  |
|          | High-vol sampler           | PM 2.5 filter samples                              | IAP        |                       |  |  |
|          | Anderson sampler           | Size- resolved PM samples                                     | IAP        |                       |  |  |
|          | ACSM                       | NR PM 1 species                                    | IAP        | Sun et al. (2012)     |  |  |
| m        | CAPS-PM- Ext    | Extinction                                                    | ΙΛΡ        | Wang at al. $(2015b)$ |  |  |
|          | (630nm)                    | Extilication                                                  | IAF        | wang et al. (2015b)   |  |  |
| .260     | SMDS 2028                  | Particle Number size                                          | ΙΛΡ        | Du at al. $(2017)$    |  |  |
| Tower ~2 | 5141 5 5756                | distribution                                                  |            | Du et al. (2017)      |  |  |
|          | Gas analyser               | $CO$ , $O_3$ and $SO_2$                                       | IAP        | Zhou et al. (2018)    |  |  |
|          | Aethalometer               | Black carbon                                                  | ΙΔΡ        | Xie et al. $(2018)$   |  |  |
|          | AE33                       | Diack carbon                                                  |            | Ale et al. (2010)     |  |  |
|          | Single particle            | Individual particles                                          | CUMTR      | Wang et al. $(2018a)$ |  |  |
|          | sampler                    | nidividual particles                                          | COMID      |                       |  |  |
|          | SNAQ boxes (x 6            | $CO NO NO_2 SO_2 PM_1$                                        |            |                       |  |  |
|          | at different               | PM 10 , PM 25                           | Cambridge  | Popoola et al. (2018) |  |  |
|          | heights)                   | - 112109 - 1122.5                                             |            |                       |  |  |
| ents     | LOPAP                      | HONO (3 min avg)                                              | Birmingham | Crilley et al. (2016) |  |  |
| eme.     |                            |                                                               |            |                       |  |  |
| easui    | Spectral radiometer        | Photolysis rates                                              | Leeds      | Bohn et al. (2016)    |  |  |
| et me    |                            | CO, NO, NO 2 , SO 2 , PM 1 , |            |                       |  |  |
| aske     | SNAQ                       | PM 10 , PM 2.5                          | Cambridge  | Popoola et al. (2018) |  |  |
| 'er b    |                            | Fluorescent biological aerosol                                |            |                       |  |  |
| l tow    | WIBS                       | particles (FBAP)                                              | IAP        | Yue et al. (2016)     |  |  |
| r and    | AE33                       | BC                                                            | IAP        | Xie et al. (2018)     |  |  |
| owe      |                            | 20                                                            |            |                       |  |  |
| L        | Los Gatos NH 3  | NH 3                                               | IAP        | Meng et al. (2018)    |  |  |
|          | Analyzer                   |                                                               |            | C                     |  |  |
|          | PAX                        | Light scattering / absorption                                 | IAP        | Xie et al. (2018)     |  |  |
| pu       |                            |                                                               |            |                       |  |  |
| grou     | High-Vol sampler           | PM 2.5 filter samples                       | Peking     |                       |  |  |
| IAP      | 4-channel sampler          | $PM_{2.5}$ filter samples                                     | Peking     |                       |  |  |

|       | -
High Vol sampler              | High time resolution PM 2.5 filter samples | York       |                     |  |
|-------|------------------------------------|--------------------------------------------------------------|------------|---------------------|--|
|       | FDMS+Thermo Sc
ientific 1405-DF | Online PM 2.5 mass conc.                          | IAP        |                     |  |
|       | Partisol sampler                   | $PM_{2.5} + PM_{2.5-10}$                                     | Birmingham | Taiwo et al. (2014) |  |
|       |                                    | Hourly elements in $PM_{2.5}$ and                            |            |                     |  |
|       | Streaker sampler                   | PM 2.5-10                                         | Birmingham | Taiwo et al. (2014) |  |
|       | Digitel High Vol                   | PM 2.5 daily                                      | IAP        |                     |  |
|       | Digitel High Vol                   | PM 1 - 3 hourly                                   | IAP        |                     |  |
|       | Andersen sampler                   | Size resolved PM                                             | IAP        |                     |  |
|       |                                    | Fluorescent biological                                       |            |                     |  |
|       | WIBS                               | particles                                                    | IAP        | Yue et al. (2016)   |  |
| p     | CAPS-NO 2               | NO 2                                              | IAP        | Ge et al. (2013)    |  |
| of/la | Aethalometer                       |                                                              |            |                     |  |
| P ro  | AE33                               | Black carbon                                                 | IAP        | Xie et al. (2018)   |  |
| IA    | CAPS-PM SSA             |                                                              |            |                     |  |
|       | (630nm)                            | Extinction, Scattering                                       | IAP        | Han et al. (2017)   |  |
|       | HR-ToF-AMS                         | NR-PM species                                                | IAP        | Sun et al. (2016)   |  |
|       |                                    | Refractory BC and coated                                     |            |                     |  |
|       | SP-AMS                             | aerosol composition                                          |            | Wang et al. (2017a) |  |
|       | Iodide FIGAERO-                    | Particle and gas phase molar                                 |            |                     |  |
|       | ToF-CIMS                           | molecule                                                     | IAP        | Zhou et al. (2018)  |  |
|       | Single particle sampler            | Individual particles                                         | CUMTB      | Wang et al. (2018)  |  |

[revised manuscript text omitted]